# In vivo single-cell high-dimensional mass cytometry analysis to track the interactions between *Klebsiella pneumoniae* and myeloid cells

**Ricardo Calderon-Gonzalez, Amy Dumigan, Joana Sá-Pessoa, Adrien Kissenpfennig, José A. Bengoechea** [ID]*

Wellcome-Wolfson Institute for Experimental Medicine, School of Medicine, Dentistry and Biomedical Sciences, Queen's University Belfast, Belfast, United Kingdom

* j.bengoechea@qub.ac.uk

**Data Availability Statement:** The authors confirm that all data underlying the findings are fully available without restriction. All relevant data are

## Abstract

In vivo single-cell approaches have transformed our understanding of the immune populations in tissues. Mass cytometry (CyTOF), that combines the resolution of mass spectrometry with the ability to conduct multiplexed measurements of cell molecules at the single cell resolution, has enabled to resolve the diversity of immune cell subsets, and their heterogeneous functionality. Here we assess the feasibility of taking CyTOF one step further to immuno profile cells while tracking their interactions with bacteria, a method we term Bac-CyTOF. We focus on the pathogen *Klebsiella pneumoniae* interrogating the pneumonia mouse model. Using Bac-CyTOF, we unveil the atlas of immune cells of mice infected with a *K. pneumoniae* hypervirulent strain. The atlas is characterized by a decrease in the populations of alveolar and monocyte-derived macrophages. Conversely, neutrophils, and inflammatory monocytes are characterized by an increase in the subpopulations expressing markers of less active cells such as the immune checkpoint PD-L1. These are the cells infected. We show that the type VI secretion system (T6SS) contributes to shape the lung immune landscape. The T6SS governs the interaction with monocytes/macrophages by shifting *Klebsiella* from alveolar macrophages to interstitial macrophages and limiting the infection of inflammatory monocytes. The lack of T6SS results in an increase of cells expressing markers of active cells, and a decrease in the subpopulations expressing PD-L1. By probing *Klebsiella*, and *Acinetobacter baumannii* strains with limited ability to survive in vivo, we uncover that a heightened recruitment of neutrophils, and relative high levels of alveolar macrophages and eosinophils and the recruitment of a characteristic subpopulation of neutrophils are features of mice clearing infections. We leverage Bac-CyTOF-generated knowledge platform to investigate the role of the DNA sensor STING in *Klebsiella* infections. *sting*⁻/⁻ infected mice present features consistent with clearing the infection including the reduced levels of PD-L1. STING absence facilitates *Klebsiella* clearance.

within the paper and its Supporting Information files.

**Funding:** This work was supported by Biotechnology and Biological Sciences Research Council (BBSRC) BB/T001976/1; BB/V007939/1 and Medical Research Council (MRC) MR/V032496/1 funds to JAB. The funders had no role in study design, data collection and analysis, decision to publish, or preparation of the manuscript.

**Competing interests:** J,A,B, declares consultancy fees from VaxDyn and GSK.

## Author summary

Host-pathogen interactions are vital to our understanding of infectious diseases, as well as its treatments and prevention. Mass cytometry and high-dimensional single-cell data analysis have provided a detail resolution of the diversity of immune cell subsets, and their heterogeneous functionality. Here we take this technology one step further to immuno profile cells in vivo while tracking their interactions with bacteria, a method we term Bac-CyTOF. Using this technology, we unveil the atlas of lung immune cells following infection with the human pathogen *Klebsiella pneumoniae*. The atlas is characterized by an increase in the populations expressing markers characteristic of less active cells such as the immune checkpoint PD-L1. The pathogen interacts with these cells. We uncover the contribution of the antimicrobial nanoweapon T6SS to shape the immune landscape, highlighting its potential in host defence. By probing two pathogens effectively cleared by mice, we reveal an immune atlas associated with clearance of infection. We leverage this knowledge platform to investigate the role of the DNA sensor STING in *Klebsiella* infections. There was no prior knowledge on the role of STING in *Klebsiella* infection biology. Our findings suggest *Klebsiella* utilizes STING signalling for its own benefit because absence of STING facilitates *Klebsiella* clearance.

## Introduction

Koch's postulates helped to revolutionize our understanding of infectious diseases by placing the microbe at the centre. This framework brought a wealth of knowledge about pathogens including the discovery of virulence factors targeting key cellular signalling pathways to promote infection. With the advent of Cellular Microbiology, pathogens and virulence factors were investigated in the context of cell lines, modelling interactions with epithelial cells lining the mucosae, and monocytes/macrophages. High-throughput screens led to the unbiased identification of bacterial and host factors governing each bacterial-cell interaction. Recent single-cell approaches have provided an unprecedent molecular resolution of the pathogen-cell interaction.

However, it is well appreciated that the pathogen-host interface in vivo cannot be recapitulated only by investigating one pathogen-cell interaction in vitro. Myeloid and non-myeloid cells interact with the pathogen resulting in the activation of several signalling pathways upon recognition of the pathogen by innate receptors. This recognition triggers the production of cytokines, chemokines and other signalling molecules that have autocrine and paracrine effects on the cells as well as act as signals to recruit immune cells. This coordinated response is essential to mount a protective response while activating pathways that may ameliorate tissue damage. Technological developments such as in vivo single-cell RNA-seq (scRNA-seq) and single-cell mass cytometry, also termed cytometry by Time-Of-Flight (hereafter termed CyTOF), now allow a system-wide view of the immune cells in vivo and their responses to pathogens. Particularly, CyTOF combines the resolution of mass spectrometry with the ability to conduct multiplexed measurements of cell molecules at the single cell resolution, allowing the detection of up to 60 markers in higher number of cells than those analyzed by scRNA-seq [1]. While the use of these technologies has transformed our understanding of the immune populations in different tissues, only recently they have been leveraged to delineate pathogen-specific immunological signatures, focusing on immune profiling but still largely ignoring the pathogen.

This work was designed to define at the cellular level the host-pathogen interface in vivo by Bacteria-CyTOF (hereafter Bac-CyTOF), an approach leveraging the power of mass cytometry to immuno profile cells while tracking their interactions with bacteria. We focused on *Klebsiella pneumoniae*, a leading cause of hospital-acquired, including ventilator-associated pneumonia, and community-acquired infections [2]. *K. pneumoniae*-triggered pulmonary infection has a high mortality rate reaching 50% even with antimicrobial therapy and may approach 100% for patients with alcoholism and diabetes [3]. In this work, we interrogated the well-established pneumonia mouse model that recapitulates key features of *Klebsiella*-induced pneumonia in humans including the massive inflammation characterized by an influx of polymorphonuclear neutrophils, and oedema [4]. At the cellular level, alveolar macrophages and inflammatory monocytes play a crucial role in host defence against *Klebsiella* [5–7] whereas interstitial macrophages promote infection [5]. Despite the recruitment of neutrophils, there are contradictory reports on whether they contribute to *Klebsiella* clearance [6,8,9]. γδ T cells, and activated NK have been reported to release cytokines necessary to activate other immune cells [10–12], and to ameliorate tissue damage upon infection by governing the rarely production of inflammatory cytokines [10,13]. Whether *Klebsiella* interacts with any of these cells is unknown.

Here we report the atlas of immune cells upon *K. pneumoniae* infection over time while tracking simultaneously the interaction of *Klebsiella* with them. By infecting with bacterial strains cleared by the host, and by challenging with a type VI secretion mutant (T6SS), we identify an immune cell milieu associated with clearance of *K. pneumoniae*. These results allowed us to predict the outcome of a *K. pneumoniae* infection in a mouse strain for which there was no previous evidence of playing any role in the host-*Klebsiella* interface, demonstrating the power of Bac-CyTOF to dissect the host-pathogen interface.

## Results

### Atlas of lung immune cells in infected mice with hypervirulent *K. pneumoniae*

To determine the immune landscape induced by *K. pneumoniae*, mice were infected intranasally with the hypervirulent *K. pneumoniae* strain CIP52.145 (hereafter Kp52145). This strain belongs to the *K. pneumoniae* KpI group and it encodes all virulence functions associated with invasive community-acquired disease in humans [14,15]. The bacterial loads in the lung increased one log every 24 h ranging from $3.5 \times 10^6 \pm 4.9 \times 10^3$ CFU/gr at 24 h to $3.4 \times 10^7 \pm 2,2 \times 10^4$ CFU/gr at 48 h to $3.2 \times 10^8 \pm 1,3 \times 10^5$ CFU/gr at 72 h post infection (S1 Fig), demonstrating that mice do not clear this infection. We used Bac-CyTOF to profile the CD45$^+$ cells of non-infected and infected mice from 24 to 72 h post infection. We tested a panel of 34 surface and intracellular markers that would enable resolution of 100 lymphoid and myeloid cell types (S1 Table). Additionally, we included in the panel an antibody anti-*Klebsiella* K2 capsule that allows tracking the association of Kp52145 with the different immune populations over time. Control experiments confirmed the specificity of the anti-capsule *Klebsiella* antibody for CyTOF (S2 Fig). The clustering algorithm PhenoGraph was used to define cell communities [16] interrogating 30,000 cells.

In infected mice, we observed a significant increase in the absolute number of neutrophils (MHC-II$^-$Ly6G$^+$Ly6C$^+$F4/80$^{-/low}$CD24$^+$) compared to non-infected mice, being the highest number at 48 h post infection ($p < 0.05$, between the numbers at 24 and 48 h; one-way ANOVA) (Fig 1A). No differences were found in the number of neutrophils at 48 h and 72 h post infection ($p > 0.05$, between the numbers at 48 and 72 h; one-way ANOVA) (Fig 1A). There was also an increase of inflammatory monocytes (SIRPa$^+$ MerTK$^-$ CD11b$^+$ MHC-II$^-$

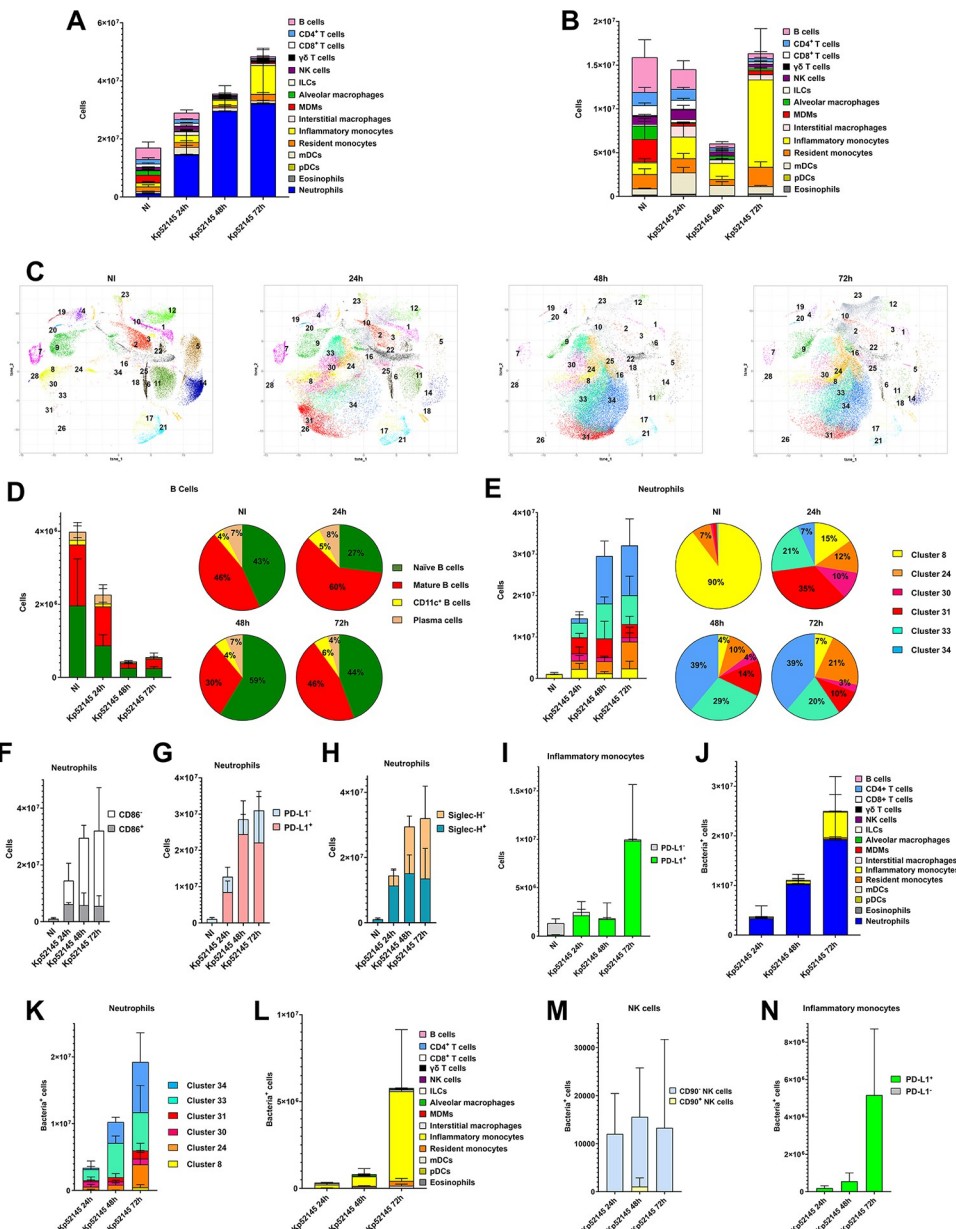

**Fig 1. Atlas of lung immune cells following infection with a virulent *K. pneumoniae* strain, Kp52145.** A. Number of immune cells in the lungs of wild-type mice non-infected (NI) or infected for 24, 48 and 72 h with Kp52145. Results (mean and the SD) are based on data from three mice per group. B. Number of non-neutrophils CD54$^+$ cells in the lungs of wild-type mice non-infected (NI) or infected for 24, 48 and 72 h with Kp52145. Results (mean and the SD) are based on data from three mice per group. C. PhenoGraph cluster analysis of immune populations in the lungs of wild-type mice non-infected (NI) or infected for 24, 48 and 72 h with Kp52145. Results are based on data from three mice per group. D. Number of cells within each of the subpopulations of B cells in the lungs of wild-type mice non-infected (NI) or infected for 24, 48 and 72 h with Kp52145. Pie charts depict the percentage of each subpopulation. Results are based on data from three mice per group. E. Number of cells per subpopulation of neutrophils in the lungs of wild-type mice non-infected (NI) or infected for 24, 48 and 72 h with Kp52145. Pie charts depict the percentage of each subpopulation. Results are based on data from three mice per group. F. Number of cells within CD86$^+$ and CD86$^-$ subpopulations of neutrophils in the lungs of wild-type mice non-infected (NI) or infected for 24, 48 and 72 h with Kp52145. Results are based on data from three mice per group. G. Number of cells within PD-L1$^+$ and PDL-L1$^-$ subpopulations of neutrophils in the lungs of wild-type mice non-infected (NI) or infected for 24, 48 and 72 h with Kp52145. Results are based on data from three mice per group. H. Number of cells within Siglec-H$^+$ and Siglec-H$^-$ subpopulations of neutrophils in the lungs of wild-type mice non-infected (NI) or infected for 24, 48 and 72 h with Kp52145. Results are based on data from three mice per group. I. Number of cells within each of the subpopulations of

inflammatory monocytes in the lungs of wild-type mice non-infected (NI) or infected for 24, 48 and 72 h with Kp52145. Results are based on data from three mice per group. J. Number of infected CD45+ cells in the lungs of infected wild-type mice for 24, 48 and 72 h with Kp52145. Results (mean and SD) are based on data from three mice per group. K. Number of infected cells within each of the subpopulations of neutrophils in the lungs of infected wild-type mice for 24, 48 and 72 h with Kp52145. Results (mean and SD) are based on data from three mice per group. L. Number of infected non-neutrophils CD45+ cells in the lungs of infected wild-type mice for 24, 48 and 72 h with Kp52145. Results (mean and SD) are based on data from three mice per group. M. Number of infected cells within each of the subpopulations of NK cells in the lungs of infected wild-type mice for 24, 48 and 72 h with Kp52145. Results are based on data from three mice per group. N. Number of infected cells within each of the subpopulations of inflammatory monocytes in the lungs of infected wild-type mice for 24, 48 and 72 h with Kp52145. Results are based on data from three mice per group.

Ly6C+ CD24-) but only at 72 h post infection (p<0.05, between the absolute numbers at 72 h versus the other two time points; one-way ANOVA) whereas at 24 and 48 h the number of these cells were not significantly different to that of non-infected mice (p>0.05, between the numbers at 24 and 48 h versus non-infected mice; one-way ANOVA) (Fig 1B). The numbers of interstitial macrophages (SIRPa+ MerTK+ CD11b+ MHC-II+) also increased in the infected mice compared to the non-infected mice (p<0.05, between the numbers at any time point post infection versus the non-infected mice; one-way ANOVA); we did not find any significant differences over time (Fig 1B). In contrast, the number of alveolar macrophages (MHC-II+Ly6-G-Ly6C-CD11blowCD11c+) and monocyte-derived alveolar macrophages (MDMs) (MerTK+ CD11c+ CD11b- MHC-II+ CCR2+) decreased in infected mice compared to non-infected ones (p<0.05, between the numbers at any time point post infection versus the non-infected mice; one-way ANOVA) (Fig 1B). At 48 and 72 h post infection, there was a significant decrease in the number of B cells (p<0.05, between the numbers at 48 and 72 h post infection versus the non-infected mice and 24 h post infection; one-way ANOVA) (Fig 1B). There were no significant differences in the numbers of other immune cells between infected and non-infected mice (Fig 1A and 1B).

For each of these 16 immune cell populations, PhenoGraph detected subpopulations only among CD4 T cells, CD8 T cells, NK cells, B cells, myeloid dendritic cells (mDCs), neutrophils, and inflammatory monocytes (Fig 1C). The markers characteristic of each of the clusters are shown in S2 Table and the heatmap of the expression of each of the markers is shown in S3 Fig. No significant differences were found in the CD8 T cells (clusters 4, 7, and 19. naïve, mature and Siglec-H+ naïve cells, respectively), CD4 T cells (clusters 1 and 6, naïve and mature cells, respectively), NK cells (clusters 17 and 21, CD90- and CD90+ cells, respectively), and mDCs (clusters 22 and 28, CD11b, and CD103 cells respectively) between the infected and the non-infected mice (S4 Fig). In contrast, we observed significant differences among infected and non-infected mice in the clusters of B cells, neutrophils, and inflammatory monocytes.

Four subpopulations of B cells were detected. No differences were found between the infected and non-infected mice in the two minor B cells clusters (clusters 3 and 23) corresponding to CD11c+ B cells, considered memory cells, and to plasma or effector B cells. (Fig 1D). At 24 h post infection there was a slight increase in the subpopulation of mature B cells (cluster 2) versus the naïve B cells (cluster 11) whereas at 48 and 72 h post infection naïve B cells, cluster 11, were the predominant B cells (Fig 1D). To reconstruct phenotypic trajectories during infection, we applied a non-linear dimensionality reduction algorithm, diffusion maps [17]. When applied to mass cytometry data generated at discrete time-points, diffusion components can be considered pseudo-time axes describing phenotypic evolution as a function of time. B cells showed a branched pattern of evolution from naïve cells, cluster 11, to plasma cells, cluster 23, and to mature and memory cells, clusters 3 and 23, respectively (S5A Fig). This pattern is characteristic of B cell maturation [18]. Altogether, this data shows that the

progression of Kp52145 infection results in a decrease in the number of B cells with populations of less differentiated B cells resembling more the B cells found in steady-state non-infected mice.

Major differences were observed in the subpopulations of neutrophils. In non-infected mice, more than 90% of the neutrophils correspond to cluster 8 (CD11b$^{low}$ Siglec-H$^+$ PD-L1$^-$ CD86$^+$) whereas this subpopulation was significantly reduced in the infected mice, particularly at 48 and 72 h post infection (Fig 1E). The other major subpopulation found in the non-infected mice was cluster 24 which can be differentiated from cluster 8 by the expression of CD86 and the relative low levels of CD44 and SIRPα (S2 Table). In contrast, at 24 post infection, the major subpopulations were clusters 31 and 33 (CD11b$^+$ Siglec-H$^+$ PD-L1$^+$) encompassing more than 50% of the neutrophils. These two clusters can be separated by the expression of CD86 (S2 Table). PD-L1 expression on neutrophils correlates with impaired antibacterial function [19–21] whereas Siglec-H is linked to negative regulation of type I IFN in myeloid cells [22]. These two subpopulations were found at 48 and 72 h post infections although the main subpopulation at these time points was cluster 34 (Siglec-H$^-$ PD-L1$^+$ CD86$^-$). This cluster is differentiated from cluster 24 by the expression of CD86 (S2 Table). Only in the infected mice, we detected neutrophils included within cluster 30 (Siglec-H$^{low}$ PD-L1$^-$ CD86$^-$) (Fig 1E), this subpopulation was reduced at 48 and 72 h post infection (p<0.05, between the numbers at 48 and 72 h post infection versus the 24 h post infection; one-way ANOVA). These cells were CD11b$^{low}$, and they were more like those neutrophils within clusters 8 and 24, predominant in the non-infected mice than to the other subpopulations of neutrophils found in the infected mice. Indeed, diffusion map analysis revealed that neutrophils started as an heterogenous population comprising clusters 8, 30 and 24, to evolve with time into two more homogenous populations, cluster 33, and clusters 34 and 31 (S5B Fig). Interestingly, the phenotypic trajectory is characterized by a progressive decrease in the number of CD86$^+$ neutrophils (Fig 1F), and an increase in the number of PD-L1$^+$ cells over time (Fig 1G), illustrating a relative increase in neutrophils with reduced ability to present antigens [23,24] and with limited antibacterial activity [19–21]. The increase of cluster 34 over time is consistent with the increase in the number of Siglec-H$^-$ cells over time (Fig 1H).

Two subpopulations of inflammatory monocytes, clusters 10 and 12, were detected in non-infected mice; these clusters can be differentiated by the expression of PD-L1 (S2 Table) being the predominant cluster 12, PD-L1$^-$ cells (Fig 1I). Remarkably, in Kp52145-infected mice we observed a switch in the subpopulation of inflammatory monocytes, being the PD-L1$^+$ subpopulation, cluster 10, the major subpopulation (Fig 1I). Cells in this cluster expressed lower levels of SIRPα, CD11b, CD44, and CD86 than cells within cluster 12 (S3 Fig); these are activation markers of monocytes/macrophages. PD-L1 inflammatory monocytes counterbalance T cell inflammatory stimuli with anti-inflammatory immune regulatory responses [25], and therefore, the PD-L1 axis is considered an important regulator of myeloid cells inflammation.

Having established the atlas of immune cells following infection, we used Bac-CyTOF to track the interactions of Kp52145 with the immune populations and subpopulations over time. PhenoGraph-based analysis showed that Kp52145 was not detected uniformly across the immune populations found in the infected mice but mostly concentrated in the clusters corresponding to neutrophils (Figs 1J and S6). The fact that we did gate for CD45$^+$ cells indicates that the bacteria detected should be at least attached to the cell surface, however the staining method also allows detecting intracellular bacteria. To clarify the localization of the detected bacteria, we carried out control experiments differentially staining the surface attached and the intracellular bacteria (S7A Fig). 80% of the bacteria detected in CD45$^+$ cells were intracellular (S7B Fig). On the contrary, most of the bacteria associated with CD45$^-$ cells were surface attached (S7B Fig). This is consistent with the limited internalization of *K. pneumoniae* by epithelial and endothelial cells, and fibroblasts [4]. When assessing the distribution of bacteria

within the CD45$^+$ cells, Kp52145 was found surface attached and intracellular in neutrophils being the predominant the latter (p<0.01, between the numbers of surface attached and intracellular bacteria; Mann-Whitney U test) (S7C Fig). In contrast, Kp52145 was predominantly surface attached to inflammatory monocytes (p<0.01, between the numbers of surface attached and intracellular bacteria; Mann-Whitney U test). In alveolar macrophages, we did not observe differences in the distribution of bacteria whereas Kp52145 was mostly surface attached in the case of resident monocytes (S7C Fig).

By labelling together surface attached bacteria and intracellular bacteria, we next questioned whether there is a uniform distribution of Kp52145 across the different subpopulations of neutrophils. At 24 h post infection, 68% of the infected neutrophils corresponded to clusters 30 and 33, whereas only 11% of the infected neutrophils were included in the cluster 31, the predominant one at this time point (Fig 1K). At 48 and 72 h post infection, there was a progressive increase in the number of infected neutrophils corresponding to cluster 34 (Fig 1K). When we segregated the subpopulations of neutrophils by markers, Kp52145 was only found within CD86$^-$ neutrophils (S8A Fig) and predominantly associated with PD-L1$^+$ cells (S8B Fig) at any time point post infection. We observed a relative increase in the number of infected Siglec-H$^-$ neutrophils (S8C Fig) consistent with the differences in the presence of subpopulations of neutrophils expressing this marker over time (Fig 1E).

Other *Klebsiella*-cell interactions were detected (Fig 1L). At 24 h post infection the main other CD45$^+$ immune populations Kp52145 positive were predominantly inflammatory monocytes, mDCs, and interstitial macrophages (Fig 1L). Other infected populations were resident monocytes, alveolar macrophages, NK cells, CD4 and CD8 T cells, and MDMs although the percentage of infected cells from the total number of non-neutrophils CD45$^+$was lower than 10% for each of them (Fig 1L). At 48 h post infection, we detected a decrease in the number of infected interstitial macrophages (3%) while the number of infected inflammatory monocytes and mDCs increased in a 62% and 11%, respectively of the non-neutrophils CD45$^+$ infected cells (Fig 1L). At 72 h post infection, in addition to neutrophils the major population of infected cells corresponded to inflammatory monocytes, reaching more than 88% of the non- neutrophils CD45$^+$ infected cells (Fig 1L). Of the two subpopulations of NK cells, Kp52145 was only found in those NK cells CD90$^+$, cluster 21 (Fig 1M). Kp52145 was only found in the PD-L1$^+$ subpopulation of inflammatory monocytes, cluster 10 (Fig 1N). Of the mDCs, only cells within cluster 22, corresponding to CD11b$^+$ cells, were infected over time.

Altogether, the immune landscape of mice infected with a *K. pneumoniae* hypervirulent strain is characterized by a decrease in the populations of cells known to control *K. pneumoniae* infections, alveolar macrophages and MDMs [5–7], whereas the other cells reported to play a crucial role in *K. pneumoniae* clearance, neutrophils, and inflammatory monocytes [6,8,9] were characterized by an increase in the subpopulations expressing markers characteristic of less active cells. Notably, these cells were those with associated bacteria. These results suggest that the progression of a hypervirulent *K. pneumoniae* infection results in a permissive lung cellular microenvironment.

## Type VI secretion system shapes the atlas of lung immune cells upon infection with virulent *K. pneumoniae*

We next aimed to decipher how hypervirulent *K. pneumoniae* shapes the atlas of lung myeloid cells. We have established that *K. pneumoniae* T6SS targets not only other bacteria but also fungi and eukaryotic cells [26,27]. Furthermore, we and others have also demonstrated that the T6SS contributes to *K. pneumoniae* infection biology [15,26–30]. Therefore, we asked whether the T6SS shapes the lung cellular environment, and whether the T6SS modulates the

interaction of *Klebsiella* with myeloid cells in vivo. We probed a *clpV* mutant [26]; the AAA$^+$ ATPase ClpV is essential for the function of *K. pneumoniae* T6SS [26]. Although at 24 h post infection the bacterial loads of *clpV*-infected mice were 2.5 x 10$^7$ ± 2.5 x 10$^4$ CFU/gr, seven times higher than that of mice infected with Kp52145, the bacterial load increased only slightly at 48 h, 4.4 x 10$^7$ 2.6 x 10$^4$ CFU/gr (S1 Fig), in contrast to the one-log increase observed in Kp52145-infected mice, suggesting that the T6SS mutant cannot flourish in the lung environment as efficiently as the wild-type strain.

At 24 h post infection, we observed higher absolute numbers of neutrophils in mice infected with the *clpV* mutant than in those infected with Kp52145 (p<0.05, between the number of neutrophils in Kp52145-infected mice versus *clpV* mutant-infected mice; one-way ANOVA) (Fig 2A). This number did not increase further at 48 h post infection in contrast to Kp52145 infected mice (p<0.05, between the number of neutrophils in Kp52145-infected mice at 24 h versus 48 h; one-way ANOVA) (Fig 2A). Further differences were noted in the population of monocytes/macrophages (Fig 2B). The number of inflammatory monocytes was higher in mice infected with the *clpV* mutant than in those infected with Kp52145 at 24 and 48 h post infection (p<0.05, between results in *clpV* mutant-infected mice and Kp52145-infected mice; one-way ANOVA) (Fig 2B) whereas the number of alveolar macrophages was higher only at 24 h post infection. At 24 h post infection, the number of interstitial macrophages was lower in mice infected with the *clpV* mutant than in those infected with Kp52145 (p<0.05, between results in *clpV* mutant-infected mice and Kp52145-infected mice; one-way ANOVA) whereas no differences were observed at 48 h post infection. Interestingly, the number of alveolar macrophages was not significantly different than those found in the non-infected mice (Fig 2B). No significant differences were noted in any of the other immune populations between *clpV*-infected and Kp52145-infected mice (Fig 2B).

Cluster analysis showed that infection with the T6SS mutant resulted in changes on the subpopulations of NK cells, B cells, neutrophils, and inflammatory monocytes. In *clpV* mutant-infected mice, we observed a switch in the predominant cluster of NK cells, being now cluster 21, encompassing activated NK cells (S3 Fig and S2 Table), the main one at 24 and 48 h post infection (53% and 51%, respectively) whereas it was cluster 17 the main one in Kp52145-infeted mice (54% at 24 h post infection, and 56% at 48 h post infection). In the subpopulations of B cells, and in contrast to Kp52145-infected mice (Fig 1C), we observed a relative increase in the number of mature B cells, cluster 2, versus naïve B cells, cluster 11, over time in *clpV* mutant-infected mice (Fig 2C). In the case of neutrophils, and similarly to Kp52145-infected mice, cluster 33 was one of the predominant clusters in *clpV* mutant-infected mice at 24 and 48 h post infection (Fig 2D). However, and in contrast to Kp52145-infected mice, clusters 30 and 24 included 42% and 46% of the neutrophils at 24 and 48 h post infection, respectively (Fig 2C) whereas in Kp5245-infected mice these subpopulations accounted for only 22 and 15% of the neutrophils at the same time points post infection (Fig 1E). These subpopulations found in *clpV* mutant-infected mice are PD-L1$^-$, and indeed there was higher number of PDL-1$^-$ neutrophils in *clpV* mutant-infected mice compared to Kp52145-infected mice, particularly at 48 h post infection (p<0.05, between results in *clpV* mutant-infected mice and Kp52145-infected mice; one-way ANOVA) (Fig 2E). A remarkable change was noted in the subpopulations of inflammatory monocytes. In the *clpV* mutant-infected mice we observed significant number of PD-L1$^-$ cells, cluster 12, at 24 h post infection which was reduced at 48 h post infection (Fig 2F). tSNE plots of PD-L1 expression further revealed the reduction of PD-L1$^+$ cells in mice infected with the *clpV* mutant, particularly at 48 h post infection, compared to mice infected with Kp52145 (S9A Fig). The more conspicuous changes were observed in the clusters corresponding to neutrophils, and alveolar macrophages. To further investigate the role of the T6SS to induce PD-L1 expression, we infected immortalized bone marrow-derived

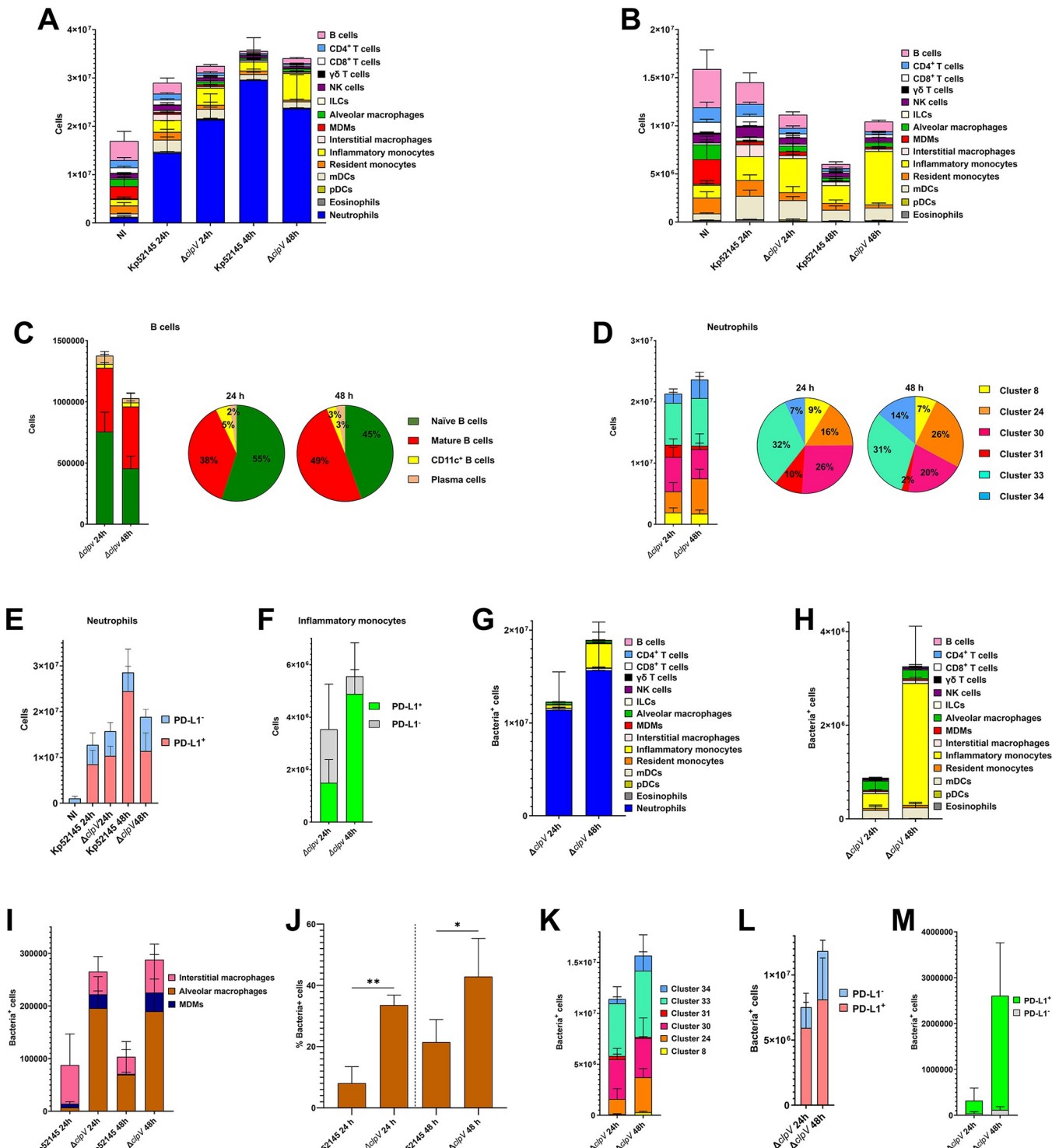

**Fig 2. Atlas of lung immune cells following infection with a T6SS *K. pneumoniae* mutant.** A. Number of immune cells in the lungs of wild-type mice non-infected (NI) or infected for 24, and 48 h with Kp52145, and the T6SS *clpV* mutant (*ΔclpV*). Results (mean and SD) are based on data from three mice per group. B. Number of non-neutrophils CD54+ cells in the lungs of wild-type mice non-infected (NI) or infected for 24, and 48 h with Kp52145, and the T6SS *clpV* mutant (*ΔclpV*). Results (mean and SD) are based on data from three mice per group. C. Number of cells within each of the subpopulations of B cells in the lungs of wild-type mice non-infected (NI) or infected for 24, and 48 h with the T6SS *clpV* mutant (*ΔclpV*). Pie charts depict the percentage of each subpopulation. Results are based on data from three mice per group. D. Number of cells within each of the subpopulations of neutrophils in the lungs of wild-type mice non-infected (NI) or infected for 24, and 48 h with the T6SS *clpV* mutant (*ΔclpV*). Pie charts depict the percentage of each subpopulation. Results are

based on data from three mice per group. E. Number of cells within the PD-L1$^+$ and PDL-L1$^-$ subpopulations of neutrophils in the lungs of wild-type mice non-infected (NI) or infected for 24, and 48 h with Kp52145, and the T6SS *clpV* mutant (Δ*clpV*). Results are based on data from three mice per group. F. Number of cells within each of the subpopulations of inflammatory monocytes in the lungs of wild-type mice infected for 24, and 48 h with Kp52145, and the T6SS *clpV* mutant (Δ*clpV*). Results are based on data from three mice per group. G. Number of infected CD45$^+$ cells in the lungs of wild-type mice infected for 24, and 48 h with the T6SS *clpV* mutant (Δ*clpV*). Results (mean and SD) are based on data from three mice per group. H. Number of infected non-neutrophils CD45$^+$ cells in the lungs of wild-type mice infected for 24, and 48 h with the T6SS *clpV* mutant (Δ*clpV*). Results (mean and SD) are based on data from three mice per group. I. Number of infected macrophages in the lungs of wild-type mice infected for 24, and 48 h with Kp52145, and the T6SS *clpV* mutant (Δ*clpV*). Results are based on data from three mice per group. J. Percentage of infected alveolar macrophages in the lungs of wild-type mice infected for 24, and 48 h with Kp52145, and the T6SS *clpV* mutant (Δ*clpV*). Results are based on data from three mice per group. **p ≤ 0.01, * p ≤ 0.05; for the indicated comparisons using one way ANOVA with Bonferroni contrast for multiple comparisons test. K. Number of infected cells within each of the subpopulations of neutrophils in the lungs of wild-type mice infected for 24, and 48 h with the T6SS *clpV* mutant (Δ*clpV*). Results (mean and SD) are based on data from three mice per group. L. Number of infected PD-L1$^+$ and PDL-L1$^-$ subpopulations of neutrophils in the lungs of wild-type mice infected for 24, and 48 h with the T6SS *clpV* mutant (Δ*clpV*). Results are based on data from three mice per group. M. Number of infected subpopulations of inflammatory monocytes in the lungs of wild-type mice challenged for 24, and 48 h with the T6SS *clpV* mutant (Δ*clpV*). Results are based on data from three mice per group.

macrophages (iBMDMs) from wild-type mice with Kp52145 and the *clpV* mutant, and the levels of PD-L1 were assessed by flow cytometry. *Klebsiella* infection increased the levels of PD-L1 in iBMDMs cells although the levels were lower in those cells infected with the *clpV* mutant than in those infected with Kp52145 (S9B Fig). Collectively, these findings indicate that the T6SS is required for *Klebsiella*-triggered upregulation of PD-L1 in vivo and in vitro.

When we tracked the interactions of the *clpV* mutant with the immune cells over time, we observed that 90% of the CD45$^+$ cells harbouring the mutant were neutrophils (Fig 2G). When considering only the non-neutrophils CD45$^+$ infected cells, at 24 h post infection the *clpV* mutant was found in alveolar macrophages, inflammatory monocytes, NK cells, MDMs, and mDCs (Fig 2H). At 48 h post infection, inflammatory monocytes accounted for 79% of the non-neutrophils CD45$^+$ infected cells whereas this percentage was only 23% in mice infected with Kp52145 (Fig 1L). Other cells infected were alveolar macrophages, mDCs, and NK cells (Fig 2H); the number of infected resident monocytes and MDMs was lower than 5% (Fig 2H). Interestingly, we noted differences in the infection of alveolar macrophages, and interstitial macrophages. At 24 h post infection, in Kp5245-infected mice interstitial macrophages were the predominant ones infected whereas in *clpV*-infected mice alveolar macrophages were the main ones infected (Fig 2I). These differences were not observed at 48 h post infection (Fig 2I). Further analysis revealed that of the total of number of alveolar macrophages only 5% of them were infected in Kp52145-mice whereas more than 25% were infected in *clpV*-infected mice (Fig 2J). The relative differences were maintained at 48 h post infection (Fig 2J). The differences in the percentage of infected alveolar macrophages between 24 and 48 h for each strain were not significantly different (Fig 2J).

We next analysed the interactions of the *clpV* mutant with the different subpopulations. Quantitative analysis showed that in the case of neutrophils the *clpV* mutant was associated with the three predominant clusters, 24, 30, and 33 with an increase in the number of infected cells encompassed in cluster 34 at 48 h post infection (Fig 2J). The mutant was mostly found in the PD-L1$^+$ subpopulations of neutrophils (Fig 2K). Similarly, in the case of inflammatory monocytes the *clpV* mutant was found selectively in those cells PD-L1$^+$ (Fig 2L), and this is despite the numbers of PD-L1$^-$ cells detected, particularly at 24 h post infection (Fig 2F).

Altogether, these results demonstrate that the lack of T6SS activity skews the atlas of immune cells towards an increase in the populations of alveolar macrophages, inflammatory monocytes, and neutrophils. Moreover, the lack of T6SS results in cells expressing activation markers, and an overall decrease in the subpopulations expressing PD-L1. Remarkably, the T6SS governs the interaction with monocytes/macrophages by shifting *Klebsiella* from the restrictive alveolar macrophages to the permissive interstitial macrophages [5] and limiting the infection of the restrictive inflammatory monocytes [7]. Collectively, this evidence highlights the contribution of *K. pneumoniae* T6SS to manipulate the lung environment.

## Atlas of lung immune cells in infected mice with *K. pneumoniae* and *Acinetobacter baumannii* strains cleared by mice

The fact that many *K. pneumoniae* clinical isolates are cleared by mice offers the opportunity to assess whether there are differences in the immune landscape induced by strains which differ in their virulence in the mouse model. We probed strain KP35 which belongs to the globally disseminated ST258 clonal group of *K. pneumoniae* carbapenem resistant strains [31]. Upon intranasal infection and, despite the initial colonization of the lung, there is a one-log decrease in bacterial loads in the lung and bronchoalveolar lavage of infected mice every 24 h [31]. At 24 h post infection the bacterial loads of KP35-inected mice were $8.5 \times 10^5 \pm 1.2 \times 10^4$ CFU/gr (S1 Fig), four time less than that of mice infected with Kp52145.

In KP35-infected mice, we also observed an increase in the number of neutrophils (Fig 3A); the absolute number was higher than that found in mice infected with Kp52145 (Fig 3A) ($p<0.01$, between the number of neutrophils in Kp52145-infected mice versus KP35-infected mice; one-way ANOVA). Further analysis of the other immune populations revealed a decrease in the number of B cells, alveolar macrophages and MDMs in KP35-infected mice ($p<0.05$, between the number of the respective cell type in non-infected mice versus

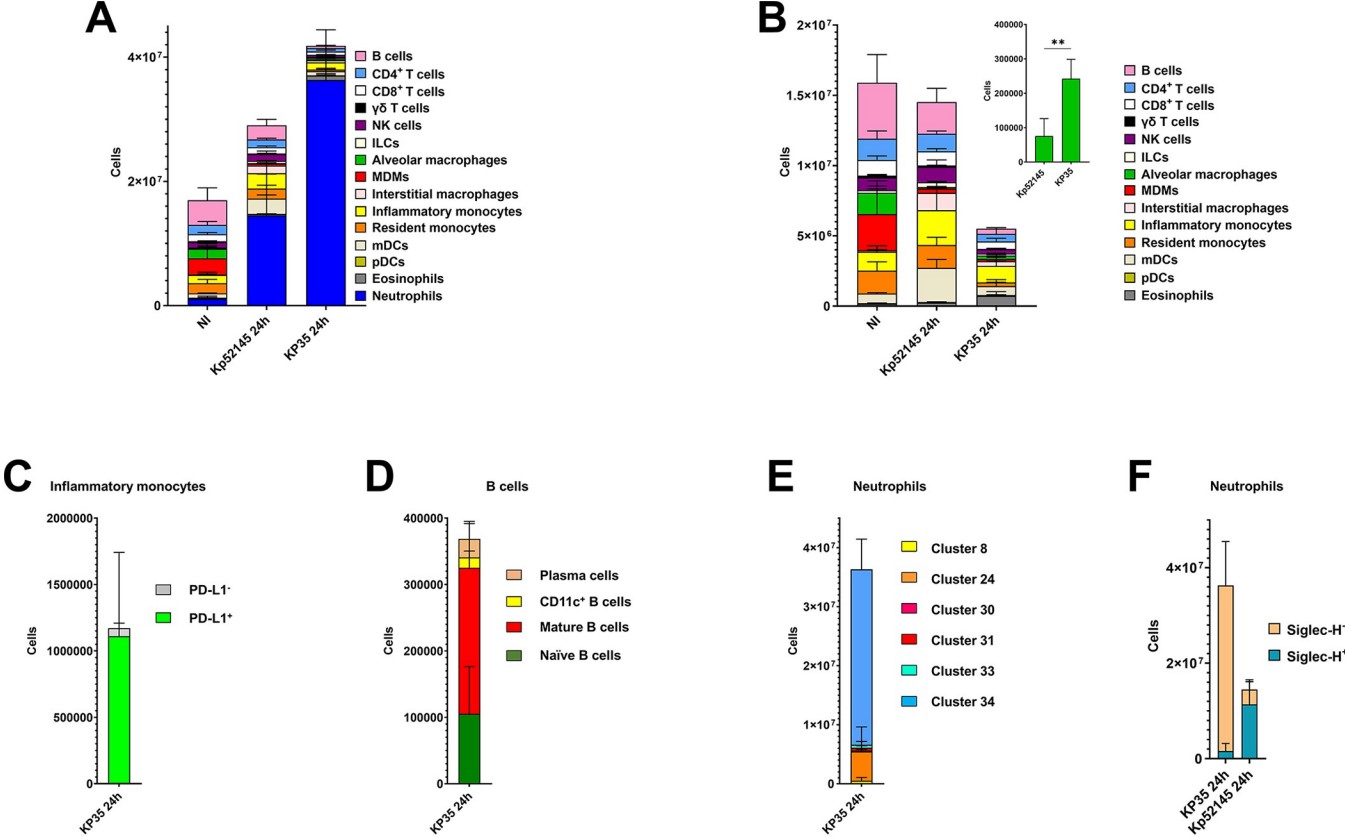

**Fig 3. Atlas of lung immune cells in mice infected with a *K. pneumoniae* strain cleared by mice, KP35.** A. Number of immune cells in the lungs of wild-type mice non-infected (NI) or infected for 24 h, with Kp52145, and KP35. Inset depicts the number of alveolar macrophages. **$p \leq 0.01$; for the indicated comparison using Mann-Whitney U test. Results (mean and SD) are based on data from three mice per group. B. Number of non-neutrophils CD54+ cells in the lungs of wild-type mice non-infected (NI) or infected for 24 h, with Kp52145, and KP35. Results (mean and SD) are based on data from three mice per group. C. Number of cells within each of the subpopulations of inflammatory monocytes in the lungs of wild-type mice infected for 24 h with KP35. D. Number of cells within each of the subpopulations of B cells in the lungs of wild-type mice infected for 24 h with KP35. E. Number of cells within each of the subpopulations of neutrophils in the lungs of wild-type mice infected for 24 h with KP35. F. Number of cells within the Siglec-H+ and Siglec-H- subpopulations of neutrophils in the lungs of wild-type mice infected for 24 h with Kp52145, and KP35. Results are based on data from three mice per group.

KP35-infected mice; one-way ANOVA) (Fig 2B), however the number of alveolar macrophages was higher in KP35-infected mice than in those infected with Kp52145 (p<0.01, between the number of alveolar macrophages in Kp52145-infected mice versus KP35-infected mice; one-way ANOVA) (Fig 3B). We also detected an increase in the number of eosinophiles, interstitial macrophages, and inflammatory monocytes compared to non-infected mice (p<0.05, between the percentage of the respective cell type in non-infected mice versus KP35-infected mice; one-way ANOVA) (Fig 3B); only the number of eosinophiles was higher in KP35-infected mice than in Kp52145-infected mice (p<0.05, between the number of eosinophils in Kp52145-infected mice versus KP35-infected mice; one-way ANOVA). There were no significant differences in the numbers of other immune cells between Kp35-infected and non-infected mice (Fig 3B).

Likewise Kp52145-infected mice, we only detected subpopulations among CD4 T cells, CD8 T cells, mDCs, NK cells, B cells, neutrophils, and inflammatory monocytes. As reported before for Kp52145-infected mice, the PD-L1$^+$ subpopulation of inflammatory monocytes, cluster 10 (Fig 3C), and the CD11b$^+$ subpopulation of mDCs (97% of the mDCs), cluster 22, were also the predominant cell populations in KP35-infected mice. In these mice, we also detected a similar ratio of CD90$^+$ (cluster 21) and CD90$^-$ (cluster 17) NK cells (46 and 54%, respectively) than those found in Kp52145-infected mice (46 and 54%, respectively) and non-infected ones (48 and 52%, respectively). However, marked differences between KP35 and Kp52145-infected mice were observed in the case of B cells, and neutrophils (Fig 3C). In the case of B cells, KP35-infected mice presented a significant increase in the number of mature B cells, cluster 2, that together with the cluster 23 corresponding to effector B cells, represent 67% of the B cells found in KP35-infected mice (Fig 3C). For neutrophils, cluster 34 was the main subpopulation in KP35-infected mice followed by cluster 24; these clusters encompassed 95% of the neutrophils (Fig 3C). Of interest, these two populations represent only 19% of the subpopulations of neutrophils in Kp52145-infected mice. Of note is the negligible number of cells within clusters 31 and 33, the predominant cell subpopulations in Kp52145-infected mice (Fig 1E). Therefore, in KP35-infected mice there was predominance of neutrophils Siglec-H$^-$ in contrast to Kp52145-infectd mice (Fig 3F).

Altogether, this data supports the notion that clearance of *K. pneumoniae* is associated with an increase recruitment of neutrophils and a relative high number of alveolar macrophages and eosinophils. Furthermore, the cellular lung microenvironment is populated by subpopulations of neutrophils, and B cells expressing markers characteristic of activated cells with antimicrobial and cytotoxic functions, resulting most likely in a restrictive lung environment for *K. pneumoniae* survival.

To establish whether this immune landscape is also found in another infection context that results in clearance of the pathogen, mice were infected with *A. baumannii* strain ATCC17978. Mice efficiently clear an infection with this *A. baumannii* strain within 24–30 h, and, therefore, Bac-CyTOF analysis was carried out 16 h post infection. At this time point, the bacterial loads in the lung were $2.6 \times 10^6 \pm 1.2 \times 10^4$ CFU/gr (S1 Fig). This infection is also associated with an increase in the absolute number of neutrophils (Fig 4A). We did not observe any decrease in the levels of alveolar macrophages (Fig 4B) in *Acinetobacter*-infected mice compared to non-infected ones whereas the number of eosinophils was higher in *Acinetobacter*-infected mice than in non-infected mice (p< 0.001 between the number of eosinophils in non-infected mice versus *Acinetobacter*-infected mice; one-way ANOVA) and not significantly different to those observed in KP35-infected mice (Fig 4B). No significant differences were noted in any of the other immune populations between non-infected and *Acinetobacter*-infected mice (Fig 4B).

Phenograph-based cluster analysis revealed that, as in the case of KP35-infected mice, cluster 34 of neutrophils was also the predominant infiltrate in *Acinetobacter*-infected mice

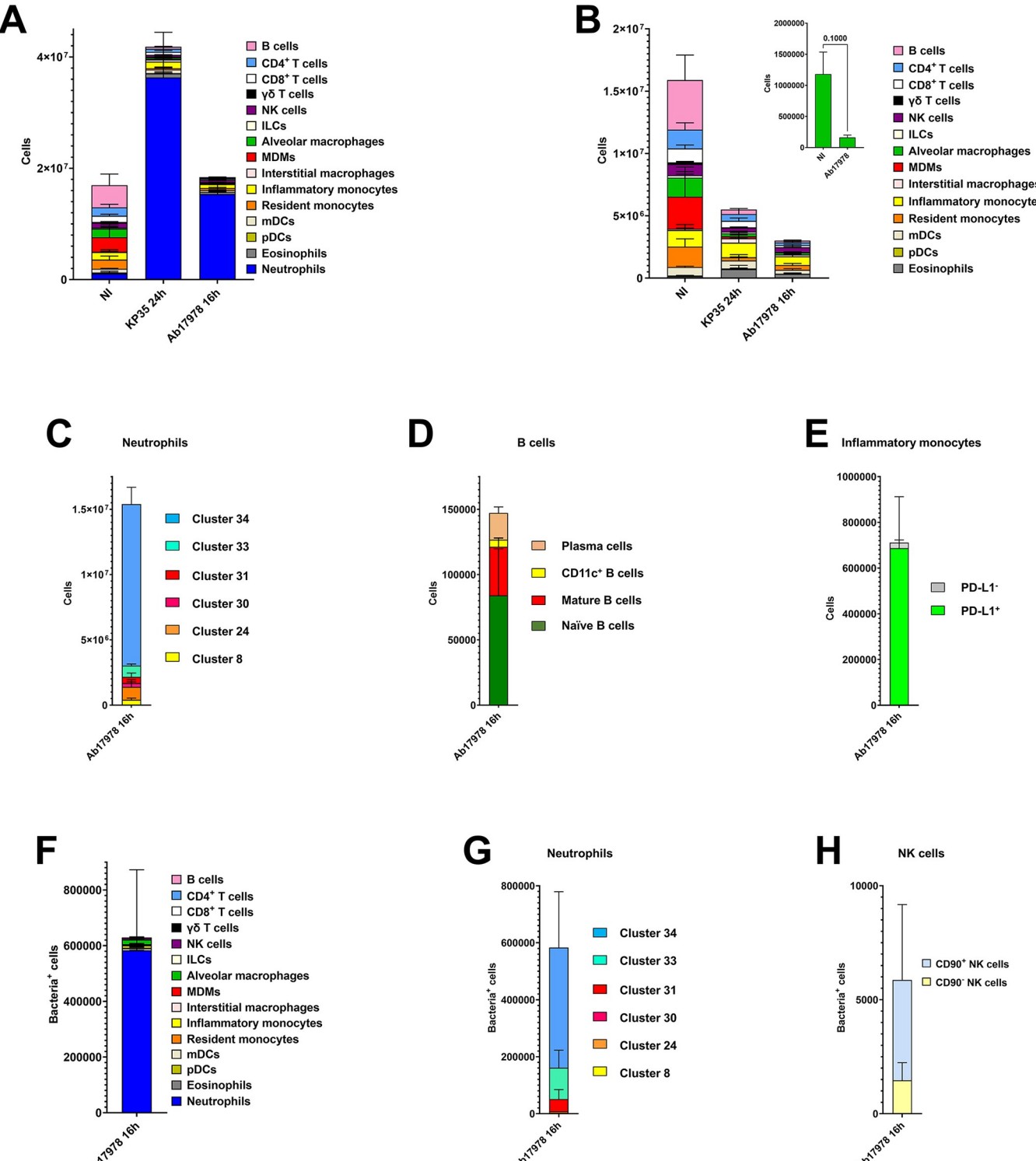

**Fig 4. Atlas of lung immune cells in mice infected with an *A. baumannii* strain cleared by mice, ATCC17978.** A. Number of immune cells in the lungs of wild-type mice non-infected (NI) or infected for 24 h with KP35, and for 16 h with *A. baumannii* strain ATCC17978. Results (mean and SD) are based on data from three mice per group. B. Number of immune cells in the lungs of wild-type mice non-infected (NI) or infected for 24 h with KP35, and for 16 h with *A. baumannii* strain ATCC17978. Inset depicts the number of alveolar macrophages. Statistical analysis done using Mann-Whitney U test. Results (mean and SD) are based on data from three mice per group. C. Number of cells within each of the subpopulations of neutrophils in the lungs of wild-type mice infected for 16 h with *A. baumannii* strain ATCC17978. Results are based on data from three mice per group. D. Number of cells within each of the subpopulations of B cells

in the lungs of wild-type mice infected for 16 h with *A. baumannii* strain ATCC17978. Results are based on data from three mice per group. E. Number of cells within each the subpopulations of inflammatory monocytes in the lungs of wild-type mice infected for 16 h with *A. baumannii* strain ATCC17978. Results are based on data from three mice per group. F. Number of infected CD45$^+$ cells in the lungs of wild-type mice infected for 16 h with *A. baumannii* strain ATCC17978. Results (mean and SD) are based on data from three mice per group. G. Number of infected cells within each of the subpopulations of neutrophils in the lungs of wild-type mice infected for 16 h with *A. baumannii* strain ATCC17978. Results (mean and SD) are based on data from three mice per group. H. Number of infected cells within each of the subpopulations of NK cells in the lungs of wild-type mice infected for 16 h with *A. baumannii* strain ATCC17978. Results (mean and SD) are based on data from three mice per group.

(Fig 4C). In the case of B cells, cluster 11, encompassing naïve B cells (S2 Table), was the predominant one, and clusters 2 and 23, encompassing mature and effector B cells, represented 40% of all B cells (Fig 4D). Like Kp52145 and KP35-infected mice, the PD-L1$^+$ subpopulation of inflammatory monocytes, cluster 10, was also the main cluster in *Acinetobacter*-infected mice (Fig 4E), indicating that this subpopulation of inflammatory monocytes is most likely linked to infection with Gram-negative bacteria irrespective of their virulence degree. Compared to non-infected mice, we detected a switch in the subpopulations of NK cells, being now cluster 21 the predominant one (55%). NK cells within this cluster expressed higher levels of CD11c, CD11b, and CD26L than cells within cluster 17 (S3 Fig). These are all markers of activation of NK cells [32]. The mDC cluster 22 was also the predominant (99%) in *Acinetobacter*-infected mice.

Control experiments confirmed the specificity of the anti-*Acinetobacter* antibody for CyTOF (S2 Fig). When we tracked the interactions of *A. baumannii* with the different immune populations, cluster analysis showed that *A. baumannii* mostly concentrated in the clusters corresponding to neutrophils and alveolar macrophages (Fig 4F). Of the CD45$^+$ infected cells, the numbers of infected mDCs, NK cells, inflammatory monocytes, interstitial macrophages, and resident monocytes were lower than 5%, (Fig 4D). No other cell types were infected. In contrast to Kp52145-infected mice, 73% of the infected neutrophils were encompassed in cluster 34 (Fig 4G), suggesting that this cluster of neutrophils may play a crucial role in the clearance of infections. There is robust evidence demonstrating that neutrophils play a critical role in host resistance to respiratory *A. baumannii* infection [33–35]. *Acinetobacter* was mostly associated with effector and mature NK cells, cluster 21 (Fig 4H), and as anticipated, only with the PD-L1$^+$ subpopulation of inflammatory monocytes, cluster 25, and with the CD103$^+$ subpopulation of mDCs, cluster 29.

Collectively, these results support the notion that a heightened recruitment of neutrophils, and relative high levels of alveolar macrophages and eosinophils are features of infected mice clearing infections by *K. pneumoniae* and *A. baumannii*. In these mice, our data also illustrates the relative increase of subpopulations of mature and effector B cells, and the predominance of the same subpopulation of neutrophils, cluster 34.

## STING signalling promotes *K. pneumoniae* infection

Bac-CyTOF results have uncovered features of infection contexts resulting in clearance of the pathogen or survival. We next sought to determine whether this knowledge can be utilized to predict the outcome of an in vivo host-*Klebsiella* interaction. We focused on STING, a DNA sensor [36], playing a major role controlling viral infections [37]. The role of STING during bacterial diseases is controversial, ranging from protective to detrimental effects for the host [38]. We have recently shown that *K. pneumoniae* activates STING signalling following infection of lung epithelial cells [27]. The role of STING on *K. pneumoniae* infection was previously unknown.

We infected *sting$^{-/-}$* mice with Kp52145, and after 24 h, we utilized Bac-CyTOF to determine the different immune populations and to track down the interactions of *K. pneumoniae* with them. Fig 5 shows that infection of *sting$^{-/-}$* mice was also associated with an increase in the

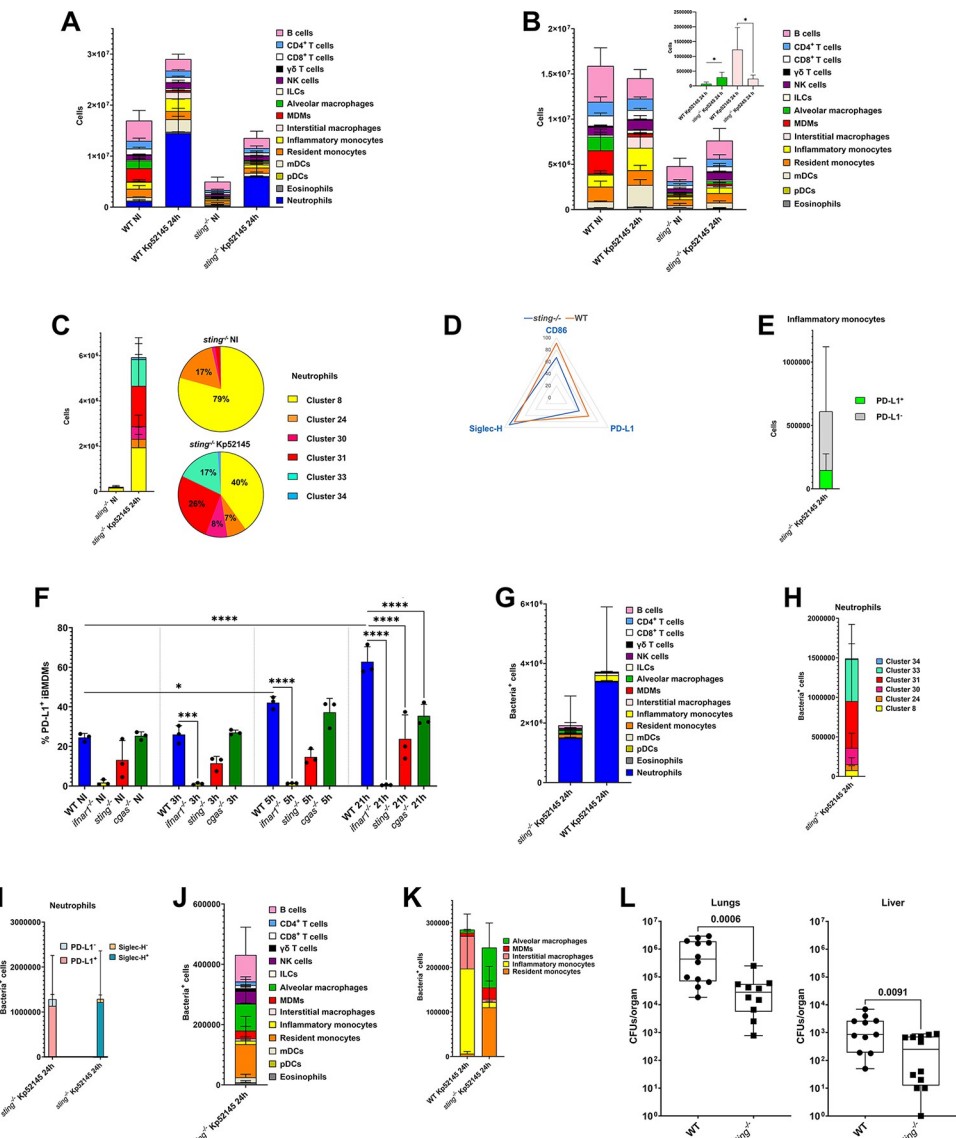

**Fig 5. STING promotes *K. pneumoniae* virulence.** A. Number of immune cells in the lungs of wild-type (WT) and *sting*⁻/⁻ mice non-infected (NI) or infected for 24 h with Kp52145. Results (mean and SD) are based on data from three mice per group. B. Number of non-neutrophils CD45⁺ cells in the lungs of wild-type (WT) and *sting*⁻/⁻ mice non-infected (NI) or infected for 24 h with Kp52145. Inset depicts the number of alveolar macrophages and interstitial macrophages. *p ≤ 0.05; for the indicated comparisons done using Mann-Whitney U test. Results (mean and SD) are based on data from three mice per group. C. Number of cells within each of the subpopulations of neutrophils in the lungs of *sting*⁻/⁻ mice non-infected (NI) or infected for 24 h with Kp52145. Pie charts depict the percentage of each subpopulation. Results are based on data from three mice per group. D. Radar plot of the markers CD86, PD-L1, and Siglec-H expressed by neutrophils in the lungs of wild-type (WT) and *sting*⁻/⁻ mice infected for 24 h with Kp52145. Results are based on data from three mice per group. E. Number of cells within each of the subpopulations of inflammatory monocytes in the lungs of *sting*⁻/⁻ mice infected for 24 h with Kp52145. Results are based on data from three mice per group. F. PD-L1 levels in iBMDMs from wild-type (WT) and *sting*⁻/⁻ mice infected with Kp52145 for the indicated time. ****p ≤ 0.0001, **p ≤ 0.01, ***p ≤ 0.001; for the indicated comparisons using one-way ANOVA with Bonferroni contrast for multiple comparisons test. G. Number of infected CD45⁺ cells in the lungs of wild-type (WT) and *sting*⁻/⁻ mice infected for 24 h with Kp52145. Results (mean and SD) are based on data from three mice per group. H. Number of infected cells within each of the subpopulations of neutrophils in the lungs of *sting*⁻/⁻ mice infected for 24 h with Kp52145. Results (mean and SD) are based on data from three mice per group. I. Number of infected PD-L1⁺, PDL-L1⁻, Siglec-H⁺, and Siglec-H⁻ cells of the subpopulations of neutrophils in the lungs of *sting*⁻/⁻ mice infected for 24 h with Kp52145. Results (mean and SD) are based on data from three mice per group. J. Number of infected non-neutrophils CD45⁺ cells in the lungs of *sting*⁻/⁻ mice infected for 24 h with Kp52145. Results (mean and SD) are based on data from three mice per group. K. Number of infected monocytes and macrophages populations in

the lungs of *sting*⁻/⁻ mice infected for 24 h with Kp52145. Results (mean and SD) are based on data from three mice per group. L. Bacterial load in the lungs and livers of infected wild-type (WT) and of *sting*⁻/⁻ for 24 h. Each dot represents a different mouse. Statistical comparisons done using the Mann-Whitney U test.

numbers of neutrophils (Fig 5A). Differences were noted in the monocytes/macrophage populations between mouse genetic backgrounds (Fig 5B). Whereas the levels of alveolar macrophages were four-fold higher in infected *sting*⁻/⁻ mice than in the wild-type ones, a five-fold decrease was observed in the levels of interstitial macrophages in infected *sting*⁻/⁻ mice compared to wild-type infected mice (Fig 5B). No significant differences were noted in any of the other immune populations between infected mouse genotypes. Collectively, absence of STING does not affect the recruitment of neutrophils and results in an increase in the number of alveolar macrophages, key cell implicated in *Klebsiella* clearance [5,6].

Cluster analysis uncovered subpopulations among CD4 T cells, CD8 T cells, mDCs, NK cells, B cells, neutrophils, and inflammatory monocytes. No differences were found in the clusters of CD4 T cells, mDC, between infected wild-type and *sting*⁻/⁻ mice (S10A and S10B Fig). In contrast, we detected a decrease in the percentage of CD8 T mature cells, cluster 7, with a concomitant increase in the percentage of naïve cells, cluster 4, in infected *sting*⁻/⁻ mice compared to wild-type ones (S10C Fig). We also observed a switch in the subpopulations of B cells in infected *sting*⁻/⁻ mice being now cluster 11 (57%) corresponding to naïve B cells (S10D Fig) the predominant one whereas cluster 2, mature B cells, was the major subpopulation in infected wild-type mice (Fig 1D). In contrast, in the case of NK cells, cluster 21, encompassing activated NK cells, was the predominant subpopulation in infected *sting*⁻/⁻ mice rather than cluster 17, corresponding to less active NK cells, detected in infected wild-type mice (S10E Fig). Remarkable differences were noted in the subpopulations of neutrophils and inflammatory monocytes. In infected *sting*⁻/⁻ mice, 40% of the neutrophils corresponded to cluster 8, characteristic of non-infected mice (Fig 5C). The other major subpopulations were clusters 31 and 33, accounting for 43% of the neutrophils. Therefore, in infected *sting*⁻/⁻ mice there was a decrease in PD-L1⁺ and CD86⁻ neutrophils compared to wild-type mice (Fig 5D). No differences were observed in the expression of Siglec-H between genotypes (Fig 5D). The switch to PDL-1⁻ subpopulation was also detected in the case of inflammatory monocytes (Fig 5E). This data is consistent with the notion that *K. pneumoniae* regulates PD-L1 in a STING-dependent manner. To investigate this further, we infected iBMDMs from wild-type and *sting*⁻/⁻ mice, and the levels of PD-L1 were assessed by flow cytometry. The levels of PD-L1 were significantly lower in Kp52145-infected *sting*⁻/⁻ iBMDMs than in Kp52145-challenegd wild-type cells (Fig 5F). To ascertain whether this phenotype is connected to DNA sensing, we infected iBMDMs from *cgas*⁻/⁻ mice. The binding of DNA to cGAS results in the production of 2′3′ cyclic GMP–AMP (cGAMP), a second messenger molecule and potent agonist of STING [39]. Infection of *cgas*⁻/⁻ also resulted in reduced levels of PD-L1 compared to wild-type cells (Fig 5F). The fact that the activation of cGAS-STING results in the production of type I IFN led us to establish whether *Klebsiella*-induced upregulation of PD-L1 was dependent on type I IFN signalling. To answer this question, we infected macrophages from *ifnar1*⁻/⁻ mice; IFNAR1 is one of the subunits of the type I IFN receptor which mediates type I IFN responses in innate and acquired immunity to infection. Kp52145 did not increase the levels of PD-L1 in *ifnar1*⁻/⁻ cells (Fig 5F), demonstrating that signalling via IFNAR1 is required for PD-L1 upregulation upon cGAS-STING activation by *Klebsiella*.

We next questioned whether absence of STING affects the interactions of *K. pneumoniae* with immune cells. Quantitative analysis of the interaction of *Klebsiella* with the different immune populations revealed that Kp52145 concentrated within neutrophils (Fig 5G). Cluster analysis showed that 53% of the infected neutrophils were encompassed in clusters 31 and 33

(Fig 5H) whereas only 9% of the infected neutrophils were within cluster 8 (Fig 5H) despite being the predominant one (Fig 5C). When the subpopulations of neutrophils were segregated by markers, Kp52145 was mostly found in PD-L1$^+$ Siglec-H$^+$ cells (Fig 5I). The other predominant CD45$^+$ immune populations Kp52145 positive were alveolar macrophages, resident monocytes, B cells, NK cells (Fig 5J). The increase interaction with alveolar macrophages led us to further investigate the distribution of Kp52145 within the populations of monocytes/macrophages. Whereas in wild-type infected mice, Kp52145 was mostly found associated with interstitial and inflammatory monocytes, alveolar macrophages and resident monocytes were the main populations associated with Kp52145 in *sting*$^{-/-}$ mice (Fig 5K). 17% and 7% of the infected non-neutrophils CD45$^+$cells corresponded to B cells and NK cells, respectively in *sting*$^{-/-}$ mice (Fig 5J). Notably, B cells were not infected in wild-type mice at 24 h post infection (Fig 1L). Cluster analysis showed that Kp52145 was associated with naïve B cells, cluster 2 (58% of the infected B cells), and mature B cells, cluster 11 (40% of the infected B cells). In the case of NK cells, Kp52145 was almost equally distributed between cluster 17 (51% of the infected NK cells) and cluster 21 (49% of the infected NK cells). Other infected cells were mDCs, inflammatory monocytes, monocyte derived macrophages, and CD4 and CD8 T cells although the percentage was lower than 5% for each of them (Fig 5J).

Altogether, these findings suggest that absence of STING results in a less immunosuppressive environment following infection with *K. pneumoniae* because of the decrease of PD-L1 expression in myeloid cells. Moreover, we observed that absence of STING skews the interaction of *K. pneumoniae* with monocytes/macrophages towards alveolar macrophages, cells we have shown restrict *K. pneumoniae* in vivo [5]. Taken together, these observations led us to predict that absence of STING should result in clearance of *K. pneumoniae* infection. To determine the ability of *sting*$^{-/-}$ mice to control bacterial growth, mice were infected intranasally. At 24 h post infection, there was a 58% reduction in bacterial load in the lungs of infected *sting*$^{-/-}$ mice compared to wild-type-infected ones (Fig 5L). Furthermore, we found a lower dissemination of Kp52145 to the liver in *sting*$^{-/-}$ mice than in wild-type ones (Fig 5L). Collectively, this evidence establishes the crucial role of STING for *K. pneumoniae* survival in vivo.

## Discusson

Here, we present a new approach, Bac-CyTOF, to dissect the host-pathogen interface at the cellular level in vivo. Bac-CyTOF allows to simultaneously immunophenotype cells, study their activation, and track the interactions of a pathogen with them. As a proof of principle, we have leveraged Bac-CyTOF to shed light into *K. pneumoniae* infection biology interrogating a pre-clinical mouse model of pneumonia. We have uncovered that hypervirulent *K. pneumoniae* induces an immunosuppressive environment characterized by PD-L1$^+$ cells, less differentiated B and NK cells, a decrease in the populations of alveolar macrophages and MDMs, and an increase in the subpopulations of neutrophils, and inflammatory monocytes expressing markers characteristic of less active cells. Notably, hypervirulent *Klebsiella* concentrated in those populations expressing markers of less active cells. This contrasts with the lung cellular environment found in mice infected with *K. pneumoniae* and *A. baumannii* strains efficiently cleared in vivo. Mechanistically, we present evidence demonstrating that *Klebsiella* T6SS plays a significant role shaping the cellular landscape of the lung, particularly the presence of subpopulations expressing PD-L1. Furthermore, we uncover that the T6SS governs the interaction of *Klebsiella* with monocytes/macrophages populations skewing the interaction from the restrictive alveolar macrophages to the permissive interstitial macrophages [5] and limiting the infection of the restrictive inflammatory monocytes [7]. Lastly, we demonstrate the role of the DNA sensor STING governing the interaction of *Klebsiella* with alveolar macrophages, B, and

NK cells, and shaping the subpopulation of neutrophils. Absence of STING facilitates the clearance of the pathogen. Collectively, our findings highlight the power of the Bac-CyTOF platform to dissect the host-pathogen interface at the single-cell level interrogating both the host and the pathogen.

CyTOF offers few advantages over classical flow cytometry to interrogate the immune landscape. Antibodies are labelled with non-biologically available metal isotopes with concise mass spectrometry parameters, overcoming the pitfalls associated with overlapping emission spectra typical of flow cytometry. Therefore, the minimal overlapping in metal signals makes theoretically possible to detect up to 100 parameters per cell. These parameters may include surface and intracellular markers, expression of cytokines, chemokines and other molecules, metabolic profile, and the activation of transcriptional factors. CyTOF then allows discerning cellular populations of interest while simultaneously interrogating facets of cellular behaviour in integrated cellular systems from a single experiment. In most settings, CyTOF is used for the detection of 42+ unique parameters rather than the 8–12 parameters that comprise a typical flow cytometry panel. A potential limitation of CyTOF is that the antibodies need to be validated for mass cytometry; however, there is an increasing number of validated antibody panels for its use in mouse and human samples. Although in this work we have focused only in CD45$^+$ cells, antibodies are available to detect epithelial and endothelial cells, and fibroblasts among other cell types, significantly expanding the portfolio of cell types that can be analysed.

Concerning the detection of bacteria, the target of the antibody should be expressed in vivo as, for example, the capsule polysaccharide of *K. pneumoniae*. Any other target can be used in principle although we strongly recommend the use of surface molecules. We envision that different antigens could be labelled allowing, for example, to follow how bacterial populations expressing different antigens interact with cells. One question is the detection of different strains from the same bacterial species which may differ in the expression of the surface antigens. To overcome this potential limitation, it is possible the use of polyclonal antisera recognizing a panel of epitopes present in several strains as we have tested in this work to detect *A. baumannii*. Of note, Bac-CyTOF allows the combination of antibodies against different bacterial species. This is of interest in the context of polymicrobial infections.

The analysis of CyTOF multidimensional data and its visualization may result cumbersome to a non-expert. However, there are excellent reviews and tutorials to facilitate the selection of the method of analysis depending on the scientific question [40], and many core facilities provide support with the initial analysis. In our team, we had previous experience only in non-multidimensional classical flow cytometry and within few weeks we set up the pipelines used in this work. Here, we have used Visualization of t-Distributed Stochastic Neighbour Embedding (t-SNE) to illustrate the structure of high-dimensional data without clustering cells into mutually exclusive groups [41]. PhenoGraph was used as an unsupervised clustering method [16]. Compare to other methods of analysis, PhenoGraph does not rely on previous knowledge of the population structure, and results in better detection of subpopulations within the same cluster of cells [42].

Using Bac-CyTOF we uncovered the expression of the immunosuppressive receptor PD-L1 in different immune populations of mice infected with virulent *K. pneumoniae*, and the concentration of *Klebsiella* in PD-L1 cells. There is extensive research on the role of the axis PD-L1/PD-1 in the immune evasion of cancer cells [43]. Therapies targeting this axis prevent T-cell exhaustion and increase a pro-inflammatory state [43]. Our findings are consistent with the notion that PD-L1 contributes to generate an immunosuppressive environment promoting *Klebsiella* infection. The role of the PD-L1/PD-1 axis in infectious diseases is still poorly characterized. Evidence mostly refers to chronic infections while the role in acute infections is less well-defined. In infections caused by human immunodeficiency virus and hepatitis B and C

viruses, the PD-L1/PD-1 axis promote infection [44–47]. Blockage of this axis enhances protective immune responses against *H. pylori* [48] whereas it worsens *M. tuberculosis* infection [49], indicating a context dependent role of PD-L1/PD-1. Our findings suggest that blockage of the PD-L1/PD-1 axis will result in diminished ability of *K. pneumoniae* to counteract the activation of protective immune responses. Future studies are warranted to confirm whether indeed this is the case.

The regulation of PD-L1 is rather complex including transcriptional and posttranscriptional mechanisms [50,51]. Here, we present evidence demonstrating that *K. pneumoniae* increases the levels of PD-L1 in an IFNAR1-cGAS-STING-dependent manner. This is consistent with the role of interferons upregulating PD-L1 [50,52–54]. To the best of our knowledge, the role of STING signalling in bacteria-induced PD-L1 has not been reported before and, only recently, STING signalling has been implicated in *Plasmodium*-triggered PD-L1 when macrophages were treated with IFNγ [55]. Interestingly, our data indicate that the T6SS governs PD-L1 levels in vivo and in vitro. Of note, we have recently demonstrated that *K. pneumoniae* encodes a T6SS effector that activates STING signalling [27] although we do not rule out that other effectors may be involved as well.

The role of type I IFN in *K. pneumoniae* infection biology is a perfect example of the host-pathogen arms race. On the one hand, type I IFN signalling is crucial to coordinate the communication between macrophages and NK cells to launch a protective immune response via IFNγ -dependent activation of alveolar macrophages [11]. On the other hand, *K. pneumoniae* leverages type I IFN signalling to skew macrophage polarization towards a singular M2-type to promote infection [5], and to increase the levels of the innate adaptor SARM1 to limit inflammation and inflammasome activation [56]. The fact that PD-L1 antagonizes type I IFN responses [57] suggests that PD-L1 activation may help *K. pneumoniae* to fine tune type I IFN signalling in vivo to survive. Interestingly, Siglec-H, found in neutrophils of *K. pneumoniae* infected mice, also acts as a negative regulator of type I IFN responses [58]. Therefore, we posit that *K. pneumoniae* leverages PD-L1 and Siglec-H to modulate type I IFN responses in vivo to promote infection. Future studies challenging tissue specific knock-outs for each of these factors will dissect their role in vivo and delineate the relative contribution of different immune cells.

Bac-CyTOF allowed us to resolve the interactions of *K. pneumoniae* with immune cells over time. While revealing the expected interactions with neutrophils and monocytes/macrophages, Bac-CyTOF also uncovered interactions with mDCs, CD4 and CD8 T cells, and NK cells. The effect of these interactions in *K. pneumoniae* infection biology is currently unknown and will be the focus of future studies. Nonetheless, we want to mark that despite the lower percentage of cells infected these could have important effects on the outcome of the infection as we have recently demonstrated for the subpopulations of lung macrophages [5].

Cluster analysis revealed the non-uniform distribution of *Klebsiella* across the different subpopulations of neutrophils predominating the infection of cells PD-L1[+] and CD86[-]. These cells are reported to have limited antibacterial activity and immunosuppressive properties [19–21]. This is consistent with the fact that despite the recruitment of neutrophils upon infection and the interaction with *Klebsiella*, these cells are not able to clear the infection. Nevertheless, we do not rule out that these subpopulations of neutrophils play an active role in *Klebsiella* strategy to generate a lung environment promoting infection. Supporting this hypothesis, accumulation of PD-L1 neutrophils in the lung after thermal injury results in increased susceptibility to *K. pneumoniae* infection [20]. Bac-CyTOF identified a population of neutrophils, cluster 34, found in wild-type mice that clear infections with *K. pneumoniae* and *A. baumannii*. This subpopulation is also found in mice infected with the hypervirulent strain Kp52145 although 24 h later than in the case of mice infected with *A. baumannii* and *K. pneumoniae* KP35, suggesting that early appearance of this subpopulation is necessary to clear the infection.

The delayed appearance of a population necessary to clear an infection is also apparent in the case of inflammatory monocytes, reaching their highest level only 72 h post infection in mice infected with Kp52145. Of note, this level was already found 24 h post infection in mice infected with *A. baumannii* and *K. pneumoniae* KP35. This observation is consistent with the evidence that the levels of these cells peaked 24 h post infection in mice clearing a *K. pneumoniae* infection [7]. The egress of these cells from the bone marrow depends on the CC-chemokine receptor 2 (CCR2) that binds CC-chemokine ligand 2 (CCL2) and CCL7 [59]. Therefore, differences in CCR2, and CCL2 and CCL7 upon infection might explain the delay recruitment of inflammatory monocytes. However, it should be noted that the recruitment of neutrophils is also CCR2-dependent [60,61], and these cells were recruited already 24 h post infection, ruling out a gross defect on egress from the bone marrow as explanation to the delayed recruitment of inflammatory monocytes. There is a gap in knowledge on whether pathogens deployed any strategy to limit the recruitment of inflammatory monocytes, making then difficult to put forward any hypothesis to explain the findings of our work. Nonetheless, our findings revealed the role of the T6SS controlling the accumulation of inflammatory monocytes, pointing out there is a T6SS effector(s), yet to be identified, governing the process of monocyte recruitment. Future studies are warranted to identify these putative anti-host effectors.

Our work emphasizes the role of *K. pneumoniae* T6SS shaping the lung cellular immune landscape. We have established its contribution to the expression of the immunosuppressive receptor PD-L1 in vitro and in vivo, and in dictating the interaction of *K. pneumoniae* with population of monocytes and macrophages controlling *Klebsiella* infections [5–7]. Additionally, recent work of the laboratory has demonstrated the action of *K. pneumoniae* T6SS on the mitochondria to promote infection [27]. Therefore, the theme taking shape is the dual role of *K. pneumoniae* T6SS as antimicrobial nanoweapon [26] including the competition with the gut microbiome [30], and as anti-host immune evasin. The diverse portfolio of T6SS effectors within *Klebsiella* strains and across the species [26] led us to postulate the existence of solely antimicrobial effectors, solely anti-host, and even some with dual role as we have recently described [26,27].

An intriguing feature of those mice clearing the infections with either KP35 or *A. baumannii* was the levels of eosinophils in the lungs in contrast to mice infected with Kp52145. While there are lung resident eosinophils, eosinophils are also recruited to the site of infection from the bone marrow [62], pointing out again that hypervirulent *K. pneumoniae* limits the influx of cells from the bone marrow. The contribution of eosinophils to clear a *K. pneumoniae* infection is unknown. In fact, it remains poorly understood the involvement of eosinophils during bacterial infections [63]. There is evidence demonstrating a direct bactericidal activity of eosinophils mediated by degranulation, and production of eosinophil extracellular traps [63]. They can promote protective pro-inflammatory responses but also contribute to pathogen-induced immunosuppression [63], highlighting the complexity of eosinophil functions in bacterial infections. Future studies are warranted to dissect the role of eosinophils in fighting a *K. pneumoniae* infection.

Another novel finding of this work is that the absence of STING facilitates the clearance of *K. pneumoniae*, suggesting that *K. pneumoniae* leverages STING to counteract host defences. The in vivo and in vitro data supports that *K. pneumoniae* exploits STING to induce PD-L1, and to govern the interaction with alveolar macrophages, resulting in a permissive environment to flourish. STING couples DNA sensing to the production of type I IFNs and related products which favour effective immune responses against viral infections. Its role during bacterial diseases is controversial, ranging from protective to detrimental effects for the host [38] which is consistent with the disparate effects of type I IFNs on bacterial infections [64]. However, the fact that abrogation of type I IFN signalling impairs the clearance of *Klebsiella* in vivo

[11] suggests that the role of STING in *K. pneumoniae* infection is more complex than only the production of type I IFNs. STING IFN-independent functions are still poorly understood. Some of them may include HMGB1 signalling, p38 MAPK signalling, NF-κB signalling, and NFAT signalling among others [65–69]. Further studies are needed to elucidate the STING-governed pathways promoting *K. pneumoniae* infection. Independently of the nature of these pathways, our findings support the notion that targeting STING is a viable strategy to favour *Klebsiella* clearance. Efforts are underway to develop pharmacological approaches to inhibit STING in a number of inflammatory diseases [70]. We propose that these drugs will show a beneficial effect to treat *Klebsiella* infections alone or as a synergistic add-on to antibiotic treatment.

## Material and methods

### Ethics statement

The experiments involving mice were approved by the Queen's University Belfast's Ethics Committee and conducted in accordance with the UK Home Office regulations (project licence PPL2910) issued by the UK Home Office.

### Bacterial strains and growth conditions

*K. pneumoniae* strain CIP52.145, Kp52145, is a clinical isolate (serotype O1:K2) previously described [15,71]. KP35 is a *K. pneumoniae* strain of the ST258 clonal group previously described [31]. The activity of the T6SS is abrogated in the Kp52145 isogenic 52145-Δ*clpV* mutant [26]. *A. baumannii* ATCC17978 strain was obtained from ATCC.

Bacteria were grown overnight in 5 mL of Miller's Luria Broth (LB) medium (Melford) at 37°C on an orbital shaker (180 rpm). Overnight bacterial cultures were refreshed 1/10 into a new tube containing 4.5 mL of fresh LB. After 2.5 h at 37°C, bacteria were pelleted (2500g, 20 min, 22°C), and resuspended in PBS to an $OD_{600}$ of 1.0 for *K. pneumoniae* (corresponding to $5x10^8$ CFUs/ml); or an $OD_{600}$ of 0.36 for *A. baumannii* (corresponding to $1x10^{10}$ CFUs/ml).

### Experimental model and subject details

C57BL/6 (Charles River) and Sting1tm1Camb (obtained from Professor Ed Lavelle -Trinity College Dublin-, originally from Dr. Laurel L. Lenz and Dr. John C. Cambier at National Jewish Health Denver, and bred at Queen's University Belfast). Mice were age and sex-matched and used between 8–12 weeks of age. Animals were randomized for interventions but researches processing the samples and analysing the data were aware which intervention group corresponded to which cohort of animals.

### Intranasal murine infection model

Infections were performed as previously described [11]. Briefly, 8- to 12-week-old mice were infected intranasally with $\sim 3 \times 10^5$ of *K. pneumoniae* strain Kp52145, $\sim 1 \times 10^8$ of *K. pneumoniae* strain KP35, or $\sim 5 \times 10^8$ of *A. baumannii* ATCC17978 in 30 μl of sterile PBS. Non-infected mice were mock infected with 30 μl sterile PBS. 3 mice were used per group. When indicated, mice were euthanized using a Schedule 1 method according to UK Home Office approved protocols. 6 hours before culling, mice were dosed intraperitoneally with 500 μg of monensin (Sigma-Aldrich) for intracellular cytokine staining. Left lung samples from infected and uninfected control mice were immersed in 1 ml of PBS for mass cytometry processing. Right lung samples from infected mice were immersed in 1 ml sterile PBS on ice and processed for quantitative bacterial culture immediately. Samples were homogenised with a Precellys

Evolution tissue homogenizer (Bertin Instruments), using 1.4 mm ceramic (zirconium oxide) beads at 4,500 rpm for 7 cycles of 10 s, with a 10-s pause between each cycle. Homogenates were serially diluted in sterile PBS and plated onto *Salmonella-Shigella* agar (Sigma-Aldrich) for *K. pneumoniae* infections or LB agar for *A. baumannii*, and the colonies were enumerated after overnight incubation at 37˚C.

## Bac-CYTOF samples preparation and analysis

**Generation of metal-labelled antibodies.** Carrier protein and glycerol-free antibodies were labelled with lanthanide isotopes using Maxpar X8 Antibody Labelling Kit or the Maxpar MCP Antibody Labelling Kit in the case of $^{110}$Cd (Standard BioTools) according to the manufacturer's instructions. Briefly, X8 or MCP9 polymers were loaded with the isotope in L-buffer, and the metal-loaded polymer purified and washed in C-buffer using an Amicon Ultra-0.5 centrifugal filter unit with 3 kDa cutoff (Millipore-Sigma). At the same time, the antibody was reduced with 4 mM tris(2-carboxyethyl)phosphine hydrochloride (TCEP) solution in R-buffer, and purified in C-buffer, using an Amicon Ultra-0.5 centrifugal filter unit with 50 kDa cut-off (Millipore-Sigma). Both the isotope-loaded polymer and the partially reduced antibody were mixed and incubated at 37˚C for 90 min. Once the incubation was completed, the conjugated antibody was washed several times with W-buffer using an Amicon Ultra-0.5 centrifugal filter unit with 50 kDa cut-off (Millipore-Sigma) and quantified using a NanoDrop spectrophotometer (280 nm). The antibody was finally resuspended in antibody stabilizer PBS supplemented with 0.05% sodium azide at a final concentration of 0.5 mg/mL and stored at 4˚C.

**Mass cytometry staining and acquisition.** Mice lungs were aseptically collected in PBS and homogenized with a handheld homogenizer. Single-cell suspensions were obtained by flushing the samples through 70 μM strainer, incubated with nuclease (Pierce). Red blood cells were lysed with ACK buffer (Gibco), and samples stained, according to manufacturer's instructions. Briefly, cell suspensions were first incubated with 1 μM of 103Rh (Standard BioTools) for live/dead discrimination, and later with antibodies against cell surface makers including, when indicated, the anti-K2 capsule antibody, prepared in Maxpar Cell Staining Buffer (CSB; Standard BioTools), for 30 min at room temperature. Cells were washed with CSB, fixed and permeabilized with Maxpar Fix I buffer (Standard BioTools) for 10 min at room temperature, washed with 2 volumes of Maxpar Perm-S buffer (Standard BioTools), and incubated with metal-labelled antibodies for intracellular markers including those against bacteria, prepared in Maxpar Perm-S buffer, for 30 min at room temperature. The list of antibodies used is shown in S1 Table. Finally, samples were washed with CSB, incubated 10 min at room temperature with a 2% paraformaldehyde solution, washed once more with CSB, and left at 4˚C in Maxpar Fix and Perm buffer (Standard BioTools) with 125 nM Cell-ID Intercalator Ir (Standard BioTools) until acquisition. Samples were acquired between 12 and 48 h after staining. Right before acquisition, cells were washed with CSB, followed by Maxpar Cell Acquisition Solution (CAS; Standard BioTools). Cells were resuspended in CAS with 1 mM EDTA to a final concentration of $1 \times 10^6$ cells/mL, flushed through a 35 μM strainer, and supplemented with 1/10 v/v EQ Four Element Calibration Beads (Standard BioTools). Mass cytometry was performed using a Helios CyTOF instrument (Standard BioTools) operated with software v7.0.8493. The CyTOF instrument was started, tuned, and cleaned according to the manufacturer's protocol, and samples acquired with an injection speed of 30 μL/min.

**Data analysis.** Data was exported as flow-cytometry FCS file format and pre-processed with CyTOF software v6.7.1014 (Standard BioTools) for normalization. Processed files were uploaded to the Cytobank platform (Beckman Coulter, https://www.cytobank.org/) for initial gating: gaussian parameters; and cells/beads, live/dead and singlets/doublets differentiation.

CD45$^+$ populations were gated and exported in FCS file format for analysis with R v4.2.3 (https://www.r-project.org/) with the integrated development environment RStudio v2023.3.1.446 (https://www.rstudio.com/), and cytofkit package (https://github.com/JinmiaoChenLab/cytofkit) for Phenograph clustering using the following parameters: 10.000 cells/sample, cytofAsinh as transformation Method, Phenograph as cluster method, k equal to 30 as Rphenograph, tsne as visualization method, a seed of 42. After clustering analysis by Phenograph, Cytofkit was used to establish diffusion maps, tool develop for single-cell analysis of differentiation data [17]. The default settings of Cytofkit were used for the analyses.

**Mammalian cells and cell culture.** Wild-type iBMDMs are a cell line derived from wild-type C57BL/6 mice and were obtained from BEI Resources (NIAID, NIH; repository number NR-9456). *Ifnar1*$^{-/-}$ iBMDMs have been previously described [56]. Additional iBMDMs were generated as previously described [72]. Briefly, tibias and femurs from male *cgas*$^{-/-}$ or *sting*$^{-/-}$ mice (C57BL/6 background, kindly donated by Andrew Bowie, Trinity College Dublin) were removed using sterile techniques, and the bone marrow was flushed with fresh medium. To obtain macrophages, cells were plated in Dulbecco's modified Eagle's medium (DMEM, high glucose, GlutaMAX Supplement) supplemented with 20% filtered L929 cell supernatant (a source of macrophage colony-stimulating factor) and maintained at 37˚C in a humidified atmosphere of 5% CO2. Medium was replaced with fresh supplemental medium after 1 day. After 5 days, BMDMs were immortalized by exposing them for 24 h to the J2 CRE virus (carrying v-myc and v-Raf/v-Mil oncogenes, kindly donated by Avinash R. Shenoy, Imperial College London). This step was repeated 2 days later (day 7), followed by continuous culture in DMEM supplemented with 20% (vol/vol) filtered L929 cell supernatant for 4 to 6 weeks. The presence of a homogeneous population of macrophages was assessed by flow cytometry using antibodies for CD11b (clone M1/70; catalog number 17-0112-82; eBioscience) and CD11c (clone N418; catalog number 48-0114-82; eBioscience).

iBMDMs were grown in DMEM (catalog number 41965; Gibco) supplemented with heat-inactivated fetal calf serum, 100 U/ml penicillin, and 0.1 mg/ml streptomycin (Gibco) at 37˚C in a humidified 5% CO2 incubator. Cells were routinely tested for *Mycoplasma* contamination.

**Infection conditions.** Overnight bacterial cultures were refreshed 1/10 into a new tube containing 4.5 ml of fresh LB. After 2.5 h at 37˚C, bacteria were pelleted (2500× g, 20 min, 22˚C), resuspended in PBS and adjusted to an optical density of 1.0 at 600 nm (5 × 108 CFU/ml). iBMDMs were seeded in 6-well plates at a density of 0.5 x 10$^6$ cells/ml in DMEM supplemented with 10% heat-inactivated FCS and allowed to adhere overnight. Infections were performed using a multiplicity of infection (MOI) of 100 bacteria per cell in a 1 ml volume. Synchronization of the infection was performed by centrifugation (200 × g for 5 min). For incubation times longer than 60 min, cells were washed and 2 ml of fresh medium containing gentamycin (100 μg/ml) was added to the wells to kill extracellular bacteria. Medium containing gentamycin was kept until the end of the experiment. Infections were performed one day after seeding the cells in the same medium used to maintain the cell line without antibiotics. Infected cells were incubated at 37˚C in a humidified 5% CO2 incubator.

**PD-L1 expression by flow cytometry.** Cells were dislodged by scraping in 0.5 ml of PBS. After washing, samples were treated with rat anti-mouse CD16/32 antibody (1:1000, clone 93, BioLegend ref. 101302) at 4˚C for 15 min to block Fc receptors. ~ 1 × 10$^6$ cells per sample were washed once and incubated with rat anti-mouse CD274/PD-L1 antibody (1:200, clone 10F.9G2, Biologend ref: 124308) for for 15 min at 4˚C. After, cells were washed with 1 ml FACS buffer (PBS with 2% FCS). Samples were then incubated in 100 μl of fixative (eBiosciences FOXp3/Transcription factor staining buffer set, ref: 00-5523-00) for 10 minutes at room temperature. Cell were then washed in 1 ml permeabilization buffer (eBiosciences

FOXp3/Transcription factor staining buffer set, ref: 00-5523-00), and stored in 100 μl of permeabilization buffer at 4°C. Right before acquisition, cells were washed once more in 1 ml of permeabilization buffer and resuspended in 100 μl of PBS containing 2.5 mM of EDTA. Flow cytometric analysis was performed using a Canto II (BD) instrument, acquiring a minimum of $1 \times 10^5$ singlets/sample. Results were exported as FCS files and analysed using FlowJo V10 (Tree Star) software.

**Statistical analysis.** Statistical analyses were performed using one-way analysis of variance (ANOVA) with Bonferroni corrections, the one-tailed t test, or, when the requirements were not met, the Mann-Whitney U test. p values of $<0.05$ were considered statistically significant. Normality and equal variance assumptions were tested with the Kolmogorov-Smirnov test and the Brown-Forsythe test, respectively. All analyses were performed using GraphPad Prism for Windows v10.0.2(232) software.

## Supporting information

**S1 Fig. Bacterial loads in the lungs of infected mice.** CFU per gr of lung of mice infected with Kp52145 for 24, 48 and 72 h, the T6SS *clpV* mutant (*ΔclpV*) for 24 andf 48 h, KP35 for 24 h, and *A. baumannii* ATCC17978 (Ab17978) for 16 h. Results (mean and SD) are based on data from three mice per group.
(TIF)

**S2 Fig. Specificity of the anti-bacteria antibodies for Bac-CyTOF.** A. Manual gating approach to identify cells positive upon incubation with the anti-*K. pneumoniae* antibody. B. Manual gating approach to identify cells positive upon incubation with the anti-*A. baumannii* antibody.
(TIF)

**S3 Fig. Heatmap of lung immune cells.** Heatmap showing relative signal intensities of the indicated markers on the populations and subpopulations found in this study. The heatmap is coloured based on signal intensity of the indicated markers. Results are based on data from three mice per group.
(TIF)

**S4 Fig. Analysis of the subpopulations of CD8 T cells, CD4 T cell, mDC, and NK cells upon infection with a virulent *K. pneumoniae* strain, Kp52145.** Radar plots show the percentage of subpopulations of A. CD8 T cells, B. CD4 T cells, C. mDCs, and D. NK cells in the lungs of wild-type mice non-infected (ni) or infected with Kp52145 for 24, 48 and 72 h.
(TIF)

**S5 Fig. Trajectories of B cells, and neutrophils upon infection with a virulent *K. pneumoniae* strain, Kp52145.** A. Diffusion map of the data set depicting B cell maturation. B. Diffusion map of the data set depicting neutrophil differentiation.
(TIF)

**S6 Fig. Analysis of the distribution of *K. pneumoniae*, strain Kp52145, in the lung immune cells.** t-SNE analysis of the populations of lung immune cells in the lungs of wild-type mice infected with Kp52145 for 24, 48 and 72 h. In red it is marked the t-SNE analysis of the *Klebsiella* marker, indicating presence of *K. pneumoniae* within the identified immune cells. Results are based on data from three mice per group.
(TIF)

**S7 Fig. Analysis of the distribution of surface attached and intracellular *K. pneumoniae*, strain Kp52145, in the lung immune cells.** A. t-SNE analysis of the populations of lung

immune cells in the lungs of wild-type mice infected with Kp52145 for 48. In red it is marked the t-SNE analysis of the *Klebsiella* marker, indicating presence of *K. pneumoniae* surface attached or intracellular within the identified immune cells. Results are based on data from three mice per group. B. Percentage of CD45$^+$ and CD45$^-$ cells with surface attached or intracellular bacteria in the lungs of wild-type mice infected with Kp52145 for 48. C. Number of cells with surface attached or intracellular bacteria in the lungs of wild-type mice infected with Kp52145 for 48. Inset depicts the number of alveolar macrophages and resident monocytes with surface attached or intracellular bacteria. Results are based on data from three mice per group.
(TIF)

**S8 Fig. Distribution of *K. pneumoniae*, strain Kp52145, in the subpopulations of neutrophils.** A. Number of infected CD86$^+$ and CD86$^-$ subpopulations of neutrophils in the lungs of wild-type mice infected for 24, 48 and 72 h with Kp52145. Results are based on data from three mice per group. B. Number of infected PD-L1$^+$ and PDL-L1$^-$ subpopulations of neutrophils in the lungs of wild-type mice infected for 24, 48 and 72 h with Kp52145. Results are based on data from three mice per group. C. Number of infected Siglec-H$^+$ and Siglec-H$^-$ subpopulations of neutrophils in the lungs of wild-type mice infected for 24, 48 and 72 h with Kp52145. Results are based on data from three mice per group.
(TIF)

**S9 Fig. T6SS controls the expression of PD-L1.** A. t-SNE analysis of the populations of lung immune cells in the lungs of wild-type mice non-infected (NI), infected with Kp52145, and the T6SS *clpV* mutant (*ΔclpV*) for 24 and 48 h. In red it is marked the t-SNE analysis of the PD-L1 marker, indicating presence of the marker within the identified immune cells. Results are based on data from three mice per group. B. PD-L1 levels in iBMDMs from wild-type mice infected with Kp52145 and the T6SS *clpV* mutant (*ΔclpV*) for the indicated time.
****$p \leq 0.0001$, **$p \leq 0.01$, *$p \leq 0.05$; for the indicated comparisons using one-way ANOVA with Bonferroni contrast for multiple comparisons test.
(TIF)

**S10 Fig. Analysis of the subpopulations of CD4 T cells, mDC, CD8 T cells, B cells and NK cells in *sting*$^{-/-}$ mice upon infection with a virulent *K. pneumoniae* strain, Kp52145.** A. Radar plots show the percentage of subpopulations of CD4 T cells in wild-type (WT) and *sting*$^{-/-}$ non-infected or infected for 24 h with Kp52145. Results are based on data from three mice per group. B. Radar plots show the percentage of subpopulations of mDCs in wild-type (WT) and *sting*$^{-/-}$ non-infected or infected for 24 h with Kp52145. Results are based on data from three mice per group. C. Radar plots show the percentage of subpopulations of CD8 T cells in wild-type (WT) and *sting*$^{-/-}$ non-infected or infected for 24 h with Kp52145. Results are based on data from three mice per group. D. Number of cells within each of the subpopulations of B cells in the lungs of *sting*$^{-/-}$ mice non-infected (NI) or infected for 24 h with Kp52145. Results are based on data from three mice per group. E. Number of cells within each of the subpopulations of NK cells in the lungs of wild-type (WT) and *sting*$^{-/-}$ non-infected (NI) or infected for 24 h with Kp52145. Results are based on data from three mice per group.
(TIF)

**S1 Table. Antibodies used for Bac-CyTOF.**
(DOCX)

**S2 Table. Immune populations.**
(DOCX)

## Acknowledgments

We thank the members of the J.A.B. laboratory for their thoughtful discussions and support with this project. We are indebted to Prof Suzana Salcedo for the kind gift of the polyclonal antiserum against *A. baumannii*. The mass cytometry equipment at Queen's University Belfast was funded by an institutional grant, and the technical support was provided by the Wellcome-Wolfson Institute for Experimental Medicine.

## Author Contributions

**Conceptualization:** Ricardo Calderon-Gonzalez, Adrien Kissenpfennig, José A. Bengoechea.

**Data curation:** Ricardo Calderon-Gonzalez, Amy Dumigan.

**Formal analysis:** Ricardo Calderon-Gonzalez, Amy Dumigan, Joana Sá-Pessoa, Adrien Kissenpfennig, José A. Bengoechea.

**Funding acquisition:** Adrien Kissenpfennig, José A. Bengoechea.

**Investigation:** Ricardo Calderon-Gonzalez, Amy Dumigan, Joana Sá-Pessoa, José A. Bengoechea.

**Methodology:** Ricardo Calderon-Gonzalez, Amy Dumigan.

**Project administration:** José A. Bengoechea.

**Supervision:** Adrien Kissenpfennig.

**Validation:** Ricardo Calderon-Gonzalez, José A. Bengoechea.

**Visualization:** Ricardo Calderon-Gonzalez.

**Writing – original draft:** Ricardo Calderon-Gonzalez, Adrien Kissenpfennig, José A. Bengoechea.

**Writing – review & editing:** Ricardo Calderon-Gonzalez, Amy Dumigan, Joana Sá-Pessoa, Adrien Kissenpfennig, José A. Bengoechea.

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
