## [Decision Letter · Decision Letter 0]

16 Jan 2024

Dear Prof. Bengoechea,

Thank you very much for submitting your manuscript "In vivo single-cell high-dimensional mass cytometry analysis to track the interaction between Klebsiella pneumoniae and myeloid cells." for consideration at PLOS Pathogens. As with all papers reviewed by the journal, your manuscript was reviewed by members of the editorial board and by several independent reviewers. In light of the reviews (below this email), we would like to invite the resubmission of a significantly-revised version that takes into account the reviewers' comments.

Thank you for submitting your work to PLOS Pathogens. Overall, the reviewers were quite enthusiastic about the findings but there are some issues to address, the main one being to determine absolute cell numbers in the data rather than relative cell proportions. Please fully address the comments in a revised version of the manuscript.

We cannot make any decision about publication until we have seen the revised manuscript and your response to the reviewers' comments. Your revised manuscript is also likely to be sent to reviewers for further evaluation.

Sincerely,

Dana J. Philpott

Academic Editor

PLOS Pathogens

David Skurnik

Section Editor

PLOS Pathogens

Kasturi Haldar

Editor-in-Chief

PLOS Pathogens

orcid.org/0000-0001-5065-158X

Michael Malim

Editor-in-Chief

PLOS Pathogens

orcid.org/0000-0002-7699-2064

Thank you for submitting your work to PLOS Pathogens. Overall, the reviewers were quite enthusiastic about the findings but there are some issues to address, the main one being to determine absolute cell numbers in the data rather than relative cell proportions. Please fully address the comments in a revised version of the manuscript.

Reviewer's Responses to Questions

**Part I - Summary**

Reviewer #1: This work by Calderon-Gonzalez et al. describes the use of CyTOF with the addition of a bacterial marker (they term Bac-CyTOF) to dissect the immune landscape of a specific bacterial infection in a given niche over time (in this case, Klebsiella pneumoniae in the lung). The authors demonstrate the relative milieu of inflammatory cells over time in the lungs of infected mice. They compare mouse lungs infected with wildtype vs. T6SS-deficient bacteria and find differences in monocytic/macrophage populations. They also look at an additional Klebsiella strain and an Acinetobacter strain and identify some differences in immune cell makeup of the lung at different timepoints. Finally, in a series of elegant experiments, the authors show that the absence of STING results in less PDL1-expressing myeloid cells and a less immunosuppressive environment in the lung. At times the writing is extremely descriptive and dense, but the authors do an admirable job with concluding paragraphs at the end of each results section highlighting the interpretation and importance of their findings. The authors also note in the Discussion that while Bac-CyTOF seems daunting at first, this technique can be powerfully applied after a short learning curve. Overall, this technology seems like a powerful tool for the field, especially when paired with multiple bacterial antibodies allowing for dissection of complicated polymicrobial infections.

Reviewer #2: In this manuscript, Calderon-Gonzalez et al perform a detailed analysis of the cell types involved in the response to Klebsiella pneumoniae (Kp) infection. The authors develop a system Bac-CyTOF, combining mass cytometry with direct labelling of Klebsiella pneumoniae to detect infected cell types to level of subclusters of immune cell populations. Overall, the atlas produced showing the kinetics of immune cell populations and changes in clusters of cells expressing different markers (i.e. PDL1) is interesting. However, there are several issues that limit the interpretation of some of the data:

Reviewer #3: This is a very interesting study that utilises single cell mass cytometry to evaluate the interactions between Klebsiella pneumoniae and host cells. The use of MS Cytof to evaluate host-pathogen interactions (which the authors coin the term Bac-CyTOF) is novel and will be of substantial value to the infection biology field. This work contributes to the knowledge on the immune evasion phenotype of k. pneumoniae.

A key strength of this work is the ability to identify the multiple immune cell types that interact and are infected by K. pneumoniae. More over the ability to hone in on specific cell populations, e.g. non-neutrophils, is informative. In addition, the availability of transgenic mouse lines from the Bowie lab (not authored) to examine specific pathways is another strength.

The identification of PDL1 activation associated with immune cell exhaustion is very interesting and will be of relevance beyond the klebsiella field.

Overall the paper shows that mice infected with their Kp strain show a decrease in populations of the cells that are required to clear infections or deactivation of cells necessary for clearance.

One issue is the biological relevance of some of the differences in cell numbers identified. While differences in the numbers of different cell types (alveolar macrophages in infected and non-infected mice; or those infected with different bacterial strains or species) have been shown and are statistically significant, it is not clear whether they are biologically significant.

It is remarkable that the comparison of the virulent Kp strain and the clpV mutant strain shows that the majority of differences are most apparent at 24 hours rather than at 48 hours, suggesting that the presence of clpV / T6SS is accelerating the effect. Can the authors comment on the kinetics of the interactions? And the fact that the differences are relatively short-term?

Two strains are compared, a virulent strain, Kp52145 and another AMR strain. While there are some differences observed between these 2 strains, some of the statements in the discussion, e.g. suggest that this is a broader effect of virulent strains generally. The authors should qualify these statements.

**Part II – Major Issues: Key Experiments Required for Acceptance**

Reviewer #1: Some additional clarifications could further strengthen this work:

-The authors describe the relative proportion of immune cells in the lung in many different figures (expressed as %CD45 cells) at many different timepoints. However, they fail to show the true number of each cell type (not just percentage) at each timepoint. For example, the total number of inflammatory cells could change over time and thus while the %CD45 may change, the true number of actual cells total may be constant (or vice versa). Providing absolute cell number (in addition to % totals) could help alleviate these concerns.

-Are the Klebsiella interactions with more effective killing cells by definition more transient, thus it appears like there are fewer by CyTOF but this is because they are doing their job. Whereas the less active cells have longer interactions with Kleb because they aren’t as good at killing?

-The addition of the bacterial marker in these CyTOF experiments seems to be the important factor. In this case, the authors use an anti-K2 antibody. Do eukaryotic cells that react with this antibody indicate that bacteria are on the surface, intracellular, or both? More discussion about what a cell positive for this marker truly means would be helpful. Also, do the authors find just bacteria in their atlas too? (K2-positive only small cells)?

-For all the different infections performed throughout the paper, the authors state the given lung titers in text, but fail to show them in graph form. Including the bacterial burdens over time in the figures would be helpful when considering the immune cellular makeups at those same timepoints.

-Line 196, does Figure 1G really show “an increase in the number of PD-L1 cells over time, or just that they are increased relative to non-infected lungs and relatively constant throughout? What about total number (not just percentage)?

Reviewer #2: 1. Throughout the manuscript, various cell types (or subclusters) are reported to be significantly reduced or increased during infection. However, given the massive influx of neutrophils in this infection model, are there actually changes in the absolute numbers of these other populations or is it only a change as a proportion of CD45+ cells?

2. As a result of the neutrophil influx after infection, many cell types (B,MDM, interstitial macrophages, CD8s etc.) appear to be <1-2% of the CD45+ population. At 30,000 cells analyzed that represents less than 300 cells being analyzed per cell type. These cells are then divided into subclusters and then further stratified by Kp+ or Kp- (i.e. NK cells 1M). This suggests that the number of cells in the subclusters must be very small for all the low abundance cell types, and even lower for the Kp+ or – grouping. These low numbers reduce confidence in interpreting the subcluster dynamics for cell types beyond neutrophils and inflammatory monocytes (Cell types which are already known to be important in response to Kp in a strain dependent manner). In line with this, plots of subclusters are shown without error bars (apart from the more abundant neutrophils in (1K).

3. Fig 1K: Do the % of infected neutrophils in different subclusters just correspond to the differing abundance of the subclusters over time or is there selectivity in the uptake? Similarly, if Figure 1L is corrected for the baseline differences in non-neutrophil cell type abundance, would that alter the interpretation?

4. Is Klebsiella bound by/associated with the different cell types or being internalized? What proportion of Klebsiella is cell associated verse free?

5. Fig 2: Why were the experiments performed only for 48 hours in this case? The authors demonstrate that there are differences in the kinetics of WT vs ClpV infections (based on CFU/gr) and the immune responses to WT Kp over the first 72 hours. The same time course should be used for the ClpV strain.

6. Fig 2: Why would ClpV be higher at 24 hours? Wouldn’t the higher bacteria load explain increased neutrophil recruitment at that timepoint rather than any difference as a result of the T6SS?

7. Fig 2AB: Given the decrease in % of neutrophils of the CD45+ population at 48hrs, wouldn’t the % of the other populations like inflammatory monocytes (the second most abundant) be larger even if there was no actual difference in the recruitment or function of the inflammatory macrophages themselves. (related to comment 1 above).

8. Lines 272-276: Cluster 17 appears to be very minor in the case of Wt Kp infection (Fig 1M). Are the cluster numbers being re-mapped with the ClpV strain? If so, the data should be analyzed together to enable direct comparison of the subclusters between the infections (as plotted in S7). If the data can't be analyzed together because it was acquired on different days, the infections should be repeated and compared directly.

9. Fig 2I: the authors show a difference in alveolar verse interstitial macrophages at 24 hours between ClpV and Wt klebsiella which normalizes by 48 hours – by which point the WT Kp have replicated to the same CFU/gr as the ClpV reach in 24 hours. This suggests that bacterial load could play an equally important role as the T6SS in determining cell populations and uptake (which seems to correlate with which cells are present in 2B)

10. Fig 5: STING experiments: Given the documented role for the microbiota to influence immune system development and function in the gut and systemically (including in the lung), are the changes in the response to Kp infection in Sting KO observed with littermate WT and STING KO mice?

11. Fig 5: STING experiments: similar to the ClpV mutant (point 5 above), why is only a single timepoint being examined? It is hard to measure clearance with a single timepoint that in other contexts is being used as a measure of initial infection. What happens at 48 and 72 hours?

12. Fig 5: The STING experiments appear to have been performed at a different time compared to the WT to which they are compared (same data is shown in Fig 5 as in Fig 1). The responses of wildtype and sting mouse lines should be compared (using littermates), during infections performed with the same inoculum and analyzed together. The changes in cell types (Fig 5A,B) and clusters are relatively minor, and could result from a different inoculum/starting bacterial load – this should be controlled for.

Reviewer #3: Differences are observed in the myeloid cell interactions between the virulent strain Kp52145 and the carbapenem resistant strain Kp35. Are these merely strain-specific differences or are there real differences between "virulent strains" and other strains? The authors should attempt to explore additional strains to verify their claims using alternative methods to determine how general these observations are.

**Part III – Minor Issues: Editorial and Data Presentation Modifications**

Reviewer #1: -I would mention that Kp52145 has markers of hypervirulence (not just say virulent) to distinguish it from the classical ST258 strain.

-Line 44, “These cells” is confusing.

-Line 49-52, confusing. Break up into two sentences, maybe?

-Line 74, delete “The”

-Line 76, what are “hubs”?

-Line 102-104 stats seem a bit antiquated. Either way, a reference is needed for those numbers.

-Line 258, “higher number” of neutrophils? I can only tell that it’s a higher percentage of neutrophils. You never shared numbers with us. This would be helpful.

-Line 298, not sure “required” is the right word because all Klebs increase the levels of PD-L1 to some degree (even though it’s less in T6SS mutants).

Reviewer #2: 13. It would be helpful for the reader if the cluster colors in pie charts consistently mapped to the cultures in the tSNE. Furthermore, in some cases, colors in legends do not match colors in pie charts

14. Fig 1D: The percentages in this panel appear to be mislabeled.

15. Throughout the figures, it would be helpful for the reader to have labels describing the cell type (immature B cell) rather than just labelling bar/pie charts with non-sequential cluster numbers

Reviewer #3: 1. The authors refer to the interaction (singular) between myeloid cells and K. pneumoniae. however it is clear that there are multiple interactions and the plural form "interactions" should be used in the title and throughout the text.

2. It is not clear how the authors establish that the cells are actually infected cells. Can the authors confirm that the host cells referred to are actually infected and not associated with the bacterial cells ? e.g. Fig 1 A and Fig 1K; Fig 2I and 2J etc.

3. In some cases, e.g. Fig 5B Lines 431 to 434, differences in levels of cells are reported which look fairly comparable. Are these biologically relevant? Can the authors report them as "fold-differences" to highlight whether they are actually different? .

4. Line 171 to 5. What is the source of the B cells being shown in Figure S4A? this section needs clarification. Which data are "these data" in line 173.

5. Figures and associated text.

While the breadth of cells that are identified is a key strength of this manuscript, the presentation of the results could be improved. It would be useful if the keys were more informative in the figures. For example B-cell clusters that are presented in Fig 1C and 2C, are identified in the text as being Naïve or mature etc, in the text, but the figure only shows cluster 11, cluster 2. The same can be said for Figs 1D & 2D and the neutrophil clusters.

The keys in figure 1 and 2 could be more informative

6. Comparisons are made between different cell populations across more than one figure. It would greatly aid the readers if the figure being compared is also identified within the main text, so that the reader doesn't have to work it out for themselves. By way of example Figure 5I is compared with Figure 3D, but it is not immediately obvious in the main text as to which of the previous figures is the one to compare. Related to this, the reader is required to flick between various figures to visualise what the authors are demonstrating, it would be useful if the WT data was shown as a comparison in figure 2, for example.

7.

Figure 4D - it appears that the A. baumannii strain is concentrated in clusters corresponding to neutrophils predominantly and the interaction with alveolar macrophages is very small in comparison (looks to be <5%).

8. Similarly the differences in alveolar macrophages in the sting-/- infected mice versus WT mice also seem to be minimal

9. There are several grammatical errors that impair smooth reading. I counted 9 grammatical errors in the abstract alone. The manuscript should be rigorously proof read by all authors before resubmission.

PLOS authors have the option to publish the peer review history of their article (what does this mean?). If published, this will include your full peer review and any attached files.

Reviewer #1: **Yes: **David Rosen

Reviewer #2: No

Reviewer #3: No
---

## [Decision Letter · Decision Letter 1]

18 Mar 2024

Dear Prof. Bengoechea,

We are pleased to inform you that your manuscript 'In vivo single-cell high-dimensional mass cytometry analysis to track the interactions between Klebsiella pneumoniae and myeloid cells.' has been provisionally accepted for publication in PLOS Pathogens.

Best regards,

Dana J. Philpott

Academic Editor

PLOS Pathogens

David Skurnik

Section Editor

PLOS Pathogens

Michael Malim

Editor-in-Chief

PLOS Pathogens

orcid.org/0000-0002-7699-2064

Congratulations on your paper!

Reviewer Comments (if any, and for reference):

Reviewer's Responses to Questions

**Part I - Summary**

Reviewer #1: My overall thoughts about this work remain largely unchanged. This work by Calderon-Gonzalez et al. describes the use of CyTOF with the addition of a bacterial marker (they term Bac-CyTOF) to dissect the immune landscape of a specific bacterial infection in a given niche over time (in this case, Klebsiella pneumoniae in the lung). The authors demonstrate the relative milieu of inflammatory cells over time in the lungs of infected mice. They compare mouse lungs infected with wildtype vs. T6SS-deficient bacteria and find differences in monocytic/macrophage populations. They also look at an additional Klebsiella strain and an Acinetobacter strain and identify some differences in immune cell makeup of the lung at different timepoints. Finally, in a series of elegant experiments, the authors show that the absence of STING results in less PDL1-expressing myeloid cells and a less immunosuppressive environment in the lung. The authors also note in the Discussion that while Bac-CyTOF seems daunting at first, this technique can be powerfully applied after a short learning curve. Overall, this technology seems like a powerful tool for the field, especially when paired with multiple bacterial antibodies allowing for dissection of complicated polymicrobial infections. The authors have also done a good job responding to my previous concerns about cell number vs. %.

Reviewer #2: The authors have nicely addressed my concerns - and those of the other reviewers.

Reviewer #3: This an interesting study that utilises single cell MS to evaluate interactions between Klebsiella pneumonia and an array of host cells. The manuscript has improved and the majority of issues have been addressed in the review. The revised manuscript will make a considerable contribution to the field.

**Part II – Major Issues: Key Experiments Required for Acceptance**

Reviewer #1: NA

Reviewer #2: (No Response)

Reviewer #3: There are no further major issues with this revised manuscript.

**Part III – Minor Issues: Editorial and Data Presentation Modifications**

Reviewer #1: NA

Reviewer #2: (No Response)

Reviewer #3: These issues have been addressed.

PLOS authors have the option to publish the peer review history of their article (what does this mean?). If published, this will include your full peer review and any attached files.

Reviewer #1: **Yes: **David A. Rosen

Reviewer #2: No

Reviewer #3: No

---

## [Editor Report · Acceptance letter]

1 Apr 2024

Dear Prof. Bengoechea,

We are delighted to inform you that your manuscript, "In vivo single-cell high-dimensional mass cytometry analysis to track the interactions between *Klebsiella pneumoniae* and myeloid cells.," has been formally accepted for publication in PLOS Pathogens.

Best regards,

Michael Malim

Editor-in-Chief

PLOS Pathogens

orcid.org/0000-0002-7699-2064